# ORDERDP: A THEORETICALLY GUARANTEED LOSS-LESS DYNAMIC DATA PRUNING FRAMEWORK

**Chenhan Jin**[1*] **Shengze Xu**[1*] **Qingsong Wang**[5] **Fan Jia**[6] **Dingshuo Chen**[4†] **Tieyong Zeng**[2,3†]

[1]The Chinese University of Hong Kong    [2]Beijing Normal-Hong Kong Baptist University
[3]Guangzhou Nanfang College    [4]Institute of Automation, Chinese Academy of Sciences
[5]Xiangtan University    [6]University of Utah

## ABSTRACT

Data pruning (DP), as an oft-stated strategy to alleviate heavy training burdens, reduces the volume of training samples according to a well-defined pruning method while striving for near-lossless performance. However, existing approaches, which commonly select highly informative samples, can lead to biased gradient estimation compared to full-dataset training. Furthermore, the analysis of this bias and its impact on final performance remains ambiguous. To address these challenges, we propose **OrderDP**, a plug-and-play framework that aims to obtain stable, unbiased, and near-lossless training acceleration with theoretical guarantees. Specifically, **OrderDP** first randomly selects a subset and then chooses the top-$q$ samples, where unbiasedness is established with respect to a surrogate loss. This ensures that **OrderDP** conducts unbiased training in terms of the surrogate objective. We further establish convergence and generalization analyses, elucidating how **OrderDP** affects optimal performance and enables well-controlled acceleration while ensuring guaranteed final performance. Empirically, we evaluate **OrderDP** against comprehensive baselines on CIFAR-10, CIFAR-100, and ImageNet-1K, demonstrating competitive accuracy, stable convergence, and exact control—all with a simpler design and faster runtime, while reducing training cost by over 40%. Delivering both strong performance and computational efficiency, our method serves as a robust and easily adaptable tool for data-efficient learning. The code is publicly available at https://github.com/shengze-xu/OrderDP.

## 1 INTRODUCTION

Neural scaling laws have revealed a consistent empirical pattern across a wide range of domains (Amari et al., 1992; Hestness et al., 2017; Kaplan et al., 2020): model performance tends to improve predictably, often as a power law (Hernandez et al., 2021; Cherti et al., 2023; Chen et al., 2023), with increased model size and the data volume. This observation has fueled a surge in computational demands and financial costs, as larger models and datasets are leveraged to push the boundary of model capabilities. In this context, data pruning (DP) has emerged as a promising strategy to alleviate training costs by selectively removing less informative samples (Killamsetty et al., 2021b; Mirzasoleiman et al., 2020; Qin et al., 2024; Raju et al., 2021), offering a pathway to enhance training efficiency without compromising model performance.

Depending on when sample selection is performed, data pruning strategies can be broadly classified into **static pruning** and **dynamic pruning**. ① Static pruning assigns an informativeness score to each training sample *before* training, typically using data influence functions (Borsos et al., 2020; Koh & Liang, 2017; Yang et al., 2022) or coreset selection strategies (Huggins et al., 2016; Campbell & Broderick, 2019; Kim et al., 2023). ② Dynamic pruning, on the other hand, performs sample selection *during* training, updating scores on-the-fly based on evolving model states or gradients (Raju et al., 2021; Qin et al., 2024; Chen et al., 2024). By continuously adapting to the training dynamics, it can better identify and retain the most influential samples at each stage, potentially yielding higher

---

[*]These authors contributed equally to this work.
[†]Corresponding author: tieyongzeng@bnbu.edu.cn, dingshuo.chen@cripac.ia.ac.cn

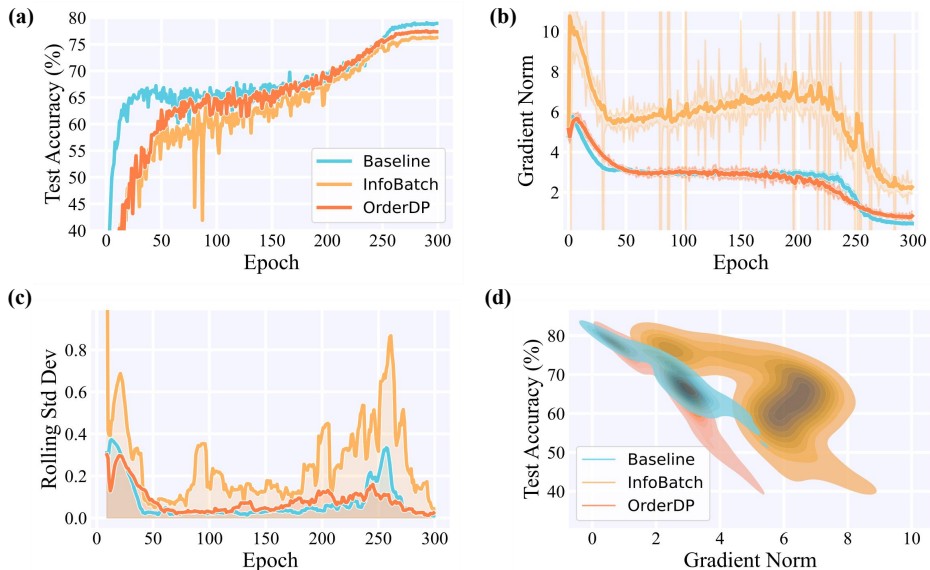

Figure 1: **Training dynamics of ResNet-18 on the CIFAR-100 under a 70% data pruning ratio.** Method comparison: full-dataset training (Baseline), representative dynamic pruning strategy (InfoBatch), and our proposed method (OrderDP). **(a-c)** Test accuracy, gradient norm (shadow area denotes standard deviation), and temporal stability of gradient norm over training epochs. **(d)** Joint distribution of test accuracy and gradient norm throughout training.

performance under constrained training budgets. A more comprehensive survey of static and dynamic pruning methods is included in Appendix B.

Within supervised learning, data pruning aims to reduce data volume without sacrificing performance, thereby achieving near-lossless[1] pruning. However, the discarded data can cause the distribution shift and bias gradient. Although selecting a portion of the discarded data randomly via calibration protocols (Ayed & Hayou, 2023) can theoretically ensure unbiasedness, finding the optimal proportion can be difficult in practice. Inspired by this method, recent dynamic methods such as InfoBatch (Qin et al., 2024) achieve this goal by rescaling the bias gradient toward the expected loss. However, when training on a specific dataset, gradient bias in both scale and direction may still arise from the discrepancy between empirical and expected loss. This bias is further amplified under extreme pruning, where large scaling factors are applied and stabilization techniques such as annealing are often required. These challenges reveal an incomplete understanding of the principles underlying "near-lossless" pruning. A critical questions arise as to *what ensures this property? how the bias should be analyzed, and whether pruning can be pushed further toward more extreme regimes?* To investigate these questions, we conduct comparative studies and report a representative result on CIFAR-100, comparing full-dataset training with InfoBatch (Qin et al., 2024) under a 70% pruning ratio to test its limits. Further experiments are presented in Appendix D, E, and we highlight several key observations as follows:

❶ *Gradient norm serves as a reliable proxy for model performance*: Under full-dataset training, test accuracy exhibits a strong linear correlation with gradient norm (Pearson's $R = -0.93$), as shown in Figure 1 (d), which suggest the magnitude of gradients is a stable and informative indicator of both training progress and generalization, which echoes prior observations in related studies (Zhao et al., 2022; Zhang et al., 2023).

❷ *Dynamic data pruning suffers from training instability*: Compared to full-dataset training, dynamic one displays pronounced fluctuations in test accuracy and volatile gradient norms. Moreover, the rolling standard deviation reveals irregular and noisy optimization dynamics, indicating reduced training stability, as shown in Figure 1 (a-c).

---

[1]Here, near-lossless means matching full-data accuracy up to normal stochastic fluctuations (typically within 0.1%) while achieving a noticeable training speedup.

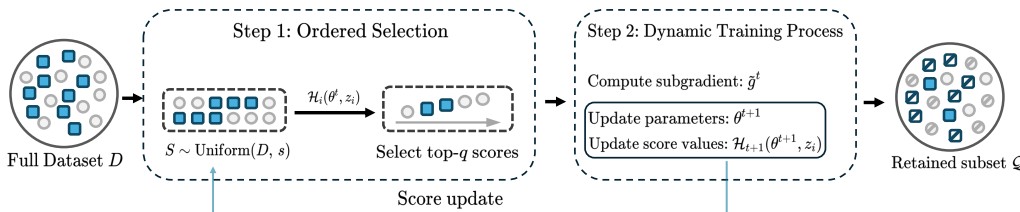

Figure 2: Illustration of the proposed **OrderDP** framework: at each iteration, a candidate batch is sampled uniformly, the top-$q$ examples are selected by score to compute a subgradient and update model parameters, and scores are refreshed only for the retained samples.

❸ *Gradient estimation under dynamic pruning is still biased*: Figure 1 (b,d) demonstrate that the dynamic method induces a noticeable shift in the overall scale of gradient norms relative to the baseline. This shift is more significant for InfoBatch, as a large scaling factor is imposed. The previously observed linear relationship between accuracy and gradient norm weakens, suggesting that it still distorts gradient estimates and introduces bias during training.

Together, these findings highlight that training instability and biased gradient estimation are two key limitations of existing dynamic DP strategies. We provide an extended empirical analysis of gradient bias in Appendix E. To this end, we take a step towards designing a DP method to mitigate both issues, thus achieving **stable, unbiased, and near-lossless** data pruning. Inspired by the recent stochastic optimization based on ordering statistics (Kawaguchi & Lu, 2020; Mehta et al., 2023), we propose a simple yet effective DP framework, **OrderDP**, which aims to obtain near-lossless pruning with improved efficiency, even at large pruning ratios. At the beginning of each epoch, we randomly sample a batch of data points from the full dataset to form a pruning candidate pool. These candidates are then ranked in descending order based on their loss values, and the Top-$q$ samples are selected as the most informative ones for model training. We formulate **OrderDP** as an optimization algorithm that minimizes a proposed surrogate loss. We theoretically establish convergence analyses using an unbiased gradient estimation of the surrogate loss. Furthermore, generalization analysis is provided in terms of the surrogate loss and expected loss, demonstrating the effectiveness of **OrderDP**.

Our approach has several desirable properties. First, **OrderDP** ensures unbiased gradient estimation and works with standard training pipelines without architectural changes or auxiliary approximations. It maintains an exactly controlled pruning ratio with rigorous theoretical guarantees on convergence and generalization. The surrogate loss fully captures the bias and enables principled, loss-aware pruning while sustaining strong stability. Empirically, we validate **OrderDP** on CIFAR-10 (Krizhevsky et al., a), CIFAR-100 (Krizhevsky et al., b), and ImageNet-1K (Deng et al., 2009). Across all benchmarks, **OrderDP** achieves near-lossless performance at moderate pruning ratios and surpasses state-of-the-art methods. On ImageNet-1K, it retains full accuracy at 40% pruning with the lowest total computation, leading to faster runtime. These results show that **OrderDP** not only sustains strong performance but also delivers superior efficiency, robustness, and a simple plug-and-play design, making it a practical solution for scalable deep learning.

## 2 METHOD

Inspired by the iterative update process of stochastic gradient descent (SGD) (Kawaguchi & Lu, 2020; Amari, 1993), we propose Ordered Data Pruning (**OrderDP**), a dynamic strategy that integrates adaptive sample selection into the SGD pipeline to reduce training cost without sacrificing accuracy. The overall framework is illustrated in Figure 2. **OrderDP** leverages a score-value mechanism to rank and retain the most informative samples at each iteration.

### 2.1 PRELIMINARIES

We begin with the standard empirical risk minimization formulation over a dataset $\mathcal{D} = \{z_i\}_{i=1}^n$:

$$\mathcal{L}(\theta) := \frac{1}{n} \sum_{i=1}^n \mathcal{L}_i(\theta, z_i) \tag{1}$$

---

**Algorithm 1:** Dynamic Training Process with **OrderDP**

---

**Input:** Initial parameters $\theta^1$, initial scores $\mathcal{H}_1(\theta^1, z_i)$ for all $i \in [n]$, learning rates $\{\eta^t\} > 0$, exploration size $s$, exploitation size $q$.

**Output:** Final parameters $\theta^T$.

1 **for** $t = 1, 2, \ldots, T$ **do**
2     // Ordered Data Pruning (**OrderDP**)
3     Sample candidate batch $S^t \subseteq D$ uniformly at random, with $|S^t| = s$.
4     Select subset $\mathcal{Q}^t \subseteq S^t$ of top-$q$ scores:
5     $$\mathcal{Q}^t \in \arg \max_{\substack{Q \subseteq S^t \\ |Q| = q}} \sum_{i \in Q} \mathcal{H}_t(\theta^t, z_i).$$
6     // Compute a subgradient
7     $\tilde{g}^t \in \partial L_{\mathcal{Q}^t}(\theta^t)$, where $L_{\mathcal{Q}^t}(\theta^t) = \dfrac{1}{q} \sum_{i \in \mathcal{Q}^t} \mathcal{L}_i(\theta^t, z_i)$.
8     // Update model parameters
9     $\theta^{t+1} \leftarrow \theta^t - \eta^t \tilde{g}^t$.
10     // Update score values
11     $\mathcal{H}_{t+1}(\theta^{t+1}, z_i) = \begin{cases} \mathcal{L}_i(\theta^t, z_i), & i \in \mathcal{Q}^t, \\ \mathcal{H}_t(\theta^t, z_i), & \text{otherwise.} \end{cases}$

---

where $\theta \in \mathbb{R}^d$ denotes the model parameters and each per-sample loss $\mathcal{L}_i(\theta, z_i): \mathbb{R}^d \to \mathbb{R}_{\geq 0}$ measures the discrepancy on example $z_i$. Solving this ERM via (mini-batch) stochastic gradient descent is at the core of most modern machine learning tasks.

**Score Value Function**. To drive dynamic pruning, we associate each sample $z_i$ with a nonnegative *score value* $\mathcal{H}_i(\theta) = \mathcal{H}_i(\theta, z_i)$, which quantifies its importance. In general, $\mathcal{H}_i$ can be any function of model state and data (e.g., gradient norm or influence measure), but we adopt the instantaneous loss $\mathcal{H}_i(\theta) = \mathcal{L}_i(\theta, z_i)$ as a simple, adaptive proxy: higher loss indicates greater need for retention. By updating scores only for samples that remain active, we avoid full-dataset recomputation each step. Concretely, if $\mathcal{Q}_t \subseteq \mathcal{D}$ denotes the set of retained examples at epoch $t$, then

$$\mathcal{H}_{t+1}(\theta^{t+1}, z_i) = \begin{cases} \mathcal{L}_i(\theta^{t+1}, z_i), & i \in \mathcal{Q}_t, \\ \mathcal{H}_t(\theta^t, z_i), & i \notin \mathcal{Q}_t. \end{cases} \quad (2)$$

This rule ensures that only the losses of selected samples are refreshed, while others retain their previous scores. Together, the ERM objective and the score value function enable dynamic data pruning: by ranking samples via $\mathcal{H}_i(\theta)$, we focus computation on the most informative subset each iteration, reducing training cost without hurting accuracy.

## 2.2 THE **OrderDP** FRAMEWORK

Building on the ERM and score-value preliminaries, Ordered Data Pruning (**OrderDP**) integrates dynamic sample selection into the SGD loop. At each iteration, a candidate batch is uniformly sampled, the top-$q$ samples are selected by score values, and a subgradient computed on this subset is used for parameter update. Scores are refreshed only for the retained samples, while others remain unchanged. The complete procedure is given in Algorithm 1.

**OrderDP** combines uniform sampling with score-based ranking to preserve diversity while focusing on informative samples, and enforces an exact prune ratio of $1 - (q/s) \cdot (s/|\mathcal{D}|)$ for predictable speed-ups. Uniform sampling ensures every sample has a non-zero chance of being selected, improving robustness (see Part 4.2, (Shah et al., 2020)) and reducing dependence on sorting. The sorting step can be reduced to $O(\log q)$ time per sample (even $O(1)$ when $q = 1$), with constant memory overhead, unlike other dynamic methods such as UCB (Raju et al., 2021) and InfoBatch (Qin et al., 2024), which require either $O(\log n)$ time or $O(n)$ storage.

# 3 THEORETICAL ANALYSIS

In this section, we show that **OrderDP** provides unbiased gradient estimates for a surrogate loss and achieves standard convex–Lipschitz convergence rates, then establish its generalization error bound via a spectral-risk analysis. Full proofs are provided in Appendix A.

## 3.1 BIAS AND CONVERGENCE ANALYSIS

In this subsection, we analyze the convergence of **OrderDP** by first capturing the bias introduced by selective pruning. Specifically, we define a surrogate loss that yields unbiased gradient updates:

$$\mathcal{L}_q(\theta) := \frac{1}{q} \sum_{j=1}^{n} \gamma_j \, \mathcal{L}_{(j)}(\theta), \quad \text{and where } \gamma_j = \sum_{l=\max\{1, s-n+j\}}^{\min\{q, j\}} \frac{\binom{j-1}{l-1} \binom{n-j}{s-l}}{\binom{n}{s}}, \tag{3}$$

where $\mathcal{L}_{(j)}(\cdot)$ is the $j$-th rank of per-sample loss and each weight $\gamma_j$ depends only on $(n, s, q)$. This construction guarantees that the gradient estimator $\tilde{g}^t$ produced by **OrderDP** is unbiased with respect to $\mathcal{L}_q$.

**Theorem 1.** *Under the definitions above, the update $\tilde{g}^t$ in Algorithm 1 satisfies*

$$\mathbb{E}[\tilde{g}^t] \notin \partial\mathcal{L}(\theta^t) \quad but \quad \mathbb{E}[\tilde{g}^t] \in \partial\mathcal{L}_q(\theta^t), \tag{4}$$

*i.e., it is an unbiased estimator of a (sub-)gradient of $\mathcal{L}_q$.*

Theorem 1 shows that the biased gradient estimation of **OrderDP** w.r.t. empirical loss $\mathcal{L}$ can be interpreted as an unbiased method for minimizing the surrogate objective $\mathcal{L}_q$. $\mathcal{L}_q$ is well-defined for any $\theta$, and **OrderDP** enjoys three key advantages: ① By choosing $s$ and $q$, the prune ratio $1 - \frac{q}{n}$ is easily adjusted; ② Computing $\gamma_j$ adds no per-epoch cost, unlike other dynamic methods requiring $O(n)$ time and memory for weight tables; ③ **OrderDP** preserves unbiased, lower-variance gradient estimates for $\mathcal{L}_q$—eliminating the need for annealing. See Appendix A.1 for the complete proof.

Another view of Theorem 1 is that the parameters $(n, s, q)$ shape the surrogate loss $\mathcal{L}_q(\theta)$ via the weights $\{\gamma_j\}$, which inherently represent the selective pruning. Inspired by the asymptotic approximation in (Kawaguchi & Lu, 2020), we obtain:

**Proposition 2.** *Denote $z = j/n$ and $\gamma(z) = \sum_{l=1}^{q} z^{l-1}(1-z)^{s-l} \frac{s!}{(l-1)!\,(s-l)!}$. Then, as $j, n \to \infty$ it holds that*

$$\lim_{j,n\to\infty,\, j/n=z} n\,\gamma_j = \gamma(z). \tag{5}$$

*Furthermore, $1 - \frac{n}{s}\,\gamma(z)$ is the cumulative distribution of $\mathrm{Beta}(z; s-q)$.*

The weight sequence $\{\gamma_j/q\}_{j=1}^{n}$ generated by **OrderDP** forms a non-uniform probability distribution (Kawaguchi & Lu, 2020; Mehta et al., 2023; Shah et al., 2020), which can be easily verified through a numerical simulation showing that $\sum_{j=1}^{n} \gamma_j/q = 1$ for given $(n, s, q)$. A non-trivial proof is also provided. For the structure of $\gamma_j$ itself, Fig 5 shows that $\gamma_j$ monotonically decays. If we fix $(s, q)$, the cliff becomes smoother and closer to $r(z)$ as $j, n$ increase. Similar observations are also found in (Kawaguchi & Lu, 2020), but we make a more general formulation of $\gamma_j$. More discussions, proof and empirical validation are deferred to Appendix A.2 and Appendix D.2.

Building on the unbiased gradient estimates of **OrderDP**, we leverage the classic mini-batch SGD analysis to obtain the following guarantee.

**Theorem 3.** *Let $(\theta^t)_{t=0}^{T}$ be the sequence generated by Algorithm 1. Suppose there exists a finite $\theta^* \in \arg\min_\theta \mathcal{L}_q(\theta)$, $\mathcal{L}_q(\theta^*) < \infty$. If each $\mathcal{L}_i(\cdot)$ is convex and $G$-Lipschitz, then*

$$\min_{0 \le t \le T} \mathbb{E}\big[\mathcal{L}_q(\theta^t) - \mathcal{L}_q(\theta^*)\big] \le \frac{\eta_{\max}\big(\|\theta^1 - \theta^*\|_2^2 + G^2 \sum_{t=1}^{T} (\eta^t)^2\big)}{2\,\eta_{\min} \sum_{t=1}^{T} \eta^t}. \tag{6}$$

This matches the standard $O(1/\sqrt{T})$ convergence rate of mini-batch SGD under the same convexity and Lipschitz assumptions, demonstrating that **OrderDP** attains identical theoretical guarantees despite pruning. In particular, choosing $\eta^t = \|\theta^1 - \theta^*\|_2/(G\sqrt{T})$ yields the error bound $(G\|\theta^1 -$

$\theta^*\|_2)/\sqrt{T}$. By using the averaged iterate $\hat{\theta}^T = (1/\sum_{t=1}^T) \sum_{t=1}^T \eta^t \theta^t$, the dependence on $\eta_{\max}$ and $\eta_{\min}$ can be removed, that is $\mathbb{E}[L_q(\bar{\theta}^T) - L_q(\theta^*)] \leq (\|\theta^1 - \theta^*\|_2^2 + G^2 \sum_{t=1}^T (\eta^t)^2)/(2\sum_{t=1}^T \eta^t)$. The full proof is provided in Appendix A.3. Empirical evidence supporting the convergence assumptions is provided in Appendix D.3.

Thus, despite pruning a large fraction of data each epoch, **OrderDP** does not slow optimization in expectation, ensuring computational savings without loss in convergence speed.

## 3.2 Generalization Analysis

Having established convergence for the surrogate loss $\mathcal{L}_q(\theta)$, we now quantify its approximation to the expected risk $\mathcal{L}(\theta^*) = \mathbb{E}_{z \sim \mathcal{D}}[\mathcal{L}(\theta^*, z)]$. Pruning creates a non-uniform sampling bias. Motivated by the 1-Wasserstein distance (Mehta et al., 2023), we rewrite $\mathcal{L}_q(\theta) = \sum_{j=1}^n \frac{\gamma_j}{q} \mathcal{L}_{(j)}(\theta)$ and $\mathcal{L}(\theta) = \sum_{j=1}^n \frac{1}{n} \mathcal{L}_{(j)}(\theta)$, thereby revealing the bias from the gap between $\{\gamma_j/q\}$ and uniform weights $\{1/n\}$. Noting that $\mathbb{E}[\mathcal{L}_q(\theta, D)] = \mathbb{E}[\sum_{j=1}^s (\hat{\gamma}_j/q) \mathcal{L}_{i_{(j)}}(\theta)]$ with $\hat{\gamma}_j = (n/s)\gamma_j$, we decompose the generalization gap $\mathbb{E}[\mathcal{L}_q(\theta, D)] - \mathcal{L}(\theta^*)$ into a bias term $\mathbb{E}[\mathcal{L}_q(\theta, D)] - \mathbb{E}[\mathcal{L}(\theta, D)]$ and a sampling error $\mathbb{E}[\mathcal{L}(\theta, D)] - \mathcal{L}(\theta^*)$, where the expectation is over the random minibatch $\{i_1, \ldots, i_s\}$.

**Theorem 4.** *(Generalization error bound). Under the same assumption of Theorem 3, the following satisfies for any $\theta^t$ in the sequence $\Theta = \{\theta\}_{t=1}^T$ generated by* **OrderDP**:

$$\mathcal{L}(\theta^*) - \mathbb{E}[\mathcal{L}_q(\theta^t, D)] \leq \underbrace{\sqrt{2}C_s B \sqrt{\frac{n-s}{s(n-1)}} - \mathcal{Q}_n(\theta^t; s, q)}_{\text{bias term}} + \underbrace{\frac{\eta_{max}(\|\theta^1 - \theta^*\|_2^2 + G^2 \sum_{t=1}^T (\eta^t)^2)}{2\eta_{min} \sum_{t=1}^T \eta^t}}_{\text{unbiased term}},$$

*where* $C_s = \sup_{t \in (0,1)} |s(t) - u(t)|^2$, $B = \inf_{\theta \in \Theta} \max_{i \in [1,n]} |\mathcal{L}_i(\theta, D_i)| < \infty$, *and* $\mathcal{Q}_n(\theta; s, q) := \inf_{\theta \in \Theta} \sum_{i=1}^n (\frac{r_i(\theta, D)}{q} - \frac{1}{n}) \mathcal{L}_i(\theta, D_i)$. *The expectation is over the random batch sampling.*

The bias term bounds the bias from selective pruning of **OrderDP**; the unbiased term is the standard optimization error, which vanishes as $T \to \infty$ with suitable $\eta^t$. The dependence on $\eta_{\max}$ and $\eta_{\min}$ can be removed by using the averaged iterate; see proof in Appendix A.4. In contrast, the value $C_s$ and $\mathcal{Q}_n(\theta; s, q)$ remain finite and quantify the deviation of $\{\gamma_j\}$ from uniformity (Mehta et al., 2023), as confirmed by simulations in Figure 6. As $q \to s$, $(r_i(\theta, D)/q - 1/n) \to 0$, so $\mathcal{Q}_n(\theta; s, q) \to 0$; and as $s \to n$, $\sqrt{2}C_s B \sqrt{\frac{n-s}{s(n-1)}} \to 0$, implying $\mathcal{L}(\theta^*) \leq \mathbb{E}[\mathcal{L}_q(\theta^t, D)]$. Thus, by minimizing $\mathcal{L}_q(\theta^t, D)$, **OrderDP** also minimizes expected generalization error. In the special case $s = q$, it reduces to standard mini-batch SGD ($r_i(\theta, D)/q = 1/n$, $C_s = 0$) and the bias vanishes. The approximation behavior characterized in Theorem 4 is further illustrated empirically in Appendix D.2.

Theorem 4 shows that **OrderDP**'s modifies the distribution shift as the gap between the surrogate loss $\mathcal{L}_q$ and the original loss $\mathcal{L}$, and the gap is fully captured by the values $C_s$ and $\mathcal{Q}_n$ of the biased term. For a high pruning ratio, i.e., small exploitation size $q$ or a small exploration size $s$ since $q \leq s$, the distribution of $\frac{\gamma_j}{q}, j \in \{1, \ldots, n\}$ exhibates a large range. This leads to a significant bias compared to the uniform distribution (which has a range of 0), a substantial discrepancy between $C_s$ and $Q_n(\theta; s, q)$, and consequently, a poor approximation. As the pruning ratio decreases (i.e., $q$ approaches $s$), the range of the $\gamma_j$ distribution narrows, and its shift from the uniform distribution diminishes and both $C_s$ and $Q_n(\theta; s, q)$ decrease, thereby improving the approximation. Specifically, when $q = s$, OrderDP reduces to standard SGD. In this case, the bias term vanishes, yielding $\mathcal{L}_q = \mathcal{L}$. We visualize the distribution shift in Figure 7.

In summary, Theorem 4 demonstrates that **OrderDP**'s generalization error comprises a vanishing optimization term and a bounded pruning bias, maintaining SGD-rate convergence while controlling dynamic pruning bias, which is consistent with the observation in Figure 1 (a-c).

---

[2] $s(t)$ and $u(t)$ refer to the probability density of the spectrum $\gamma_j$ distribution and uniform distribution on $(0, 1)$. Details can be found in (Mehta et al., 2023).

## 4 EXPERIMENTAL SETTINGS

### 4.1 DATASETS AND TASKS

To comprehensively validate the effectiveness of our proposed **OrderDP**, we conduct experiments on a range of image classification benchmarks: CIFAR-10 and CIFAR-100 (Krizhevsky et al., a;b), ImageNet-1K (Deng et al., 2009).

CIFAR datasets comprise $32 \times 32$ color images across 10 and 100 categories, respectively. Each split includes 50,000 training and 10,000 test samples, providing balanced classification evaluation. ImageNet-1K, as a 1,000-class subset of ImageNet-21k, contains 1,281,167 training images and 50,000 validation images, spanning a variety of object categories.

### 4.2 IMPLEMENTATION DETAILS

In this section, we provide a succinct overview of the implementation details for our experiments, including backbone models and training details.

**Backbone models.** For classification, we train ResNet-18 and ResNet-50 (He et al., 2016) on CIFAR-10/100 and ImageNet-1K.

**Training Details.** For **OrderDP**, an exploration ratio of 0.5 related to $s$ (i.e., $s/|\mathcal{D}| = 0.5$) and an exploitation ratio of 0.6 related to $q$ (i.e., $q/s = 0.6$) are used by default when no other values are specified. All models are trained with the OneCycle scheduler, which employs cosine annealing, using SGD with a momentum of 0.9 and a weight decay of $5 \times 10^{-4}$. Images are augmented through normalization, random cropping, and horizontal flipping unless stated otherwise. The implementation is based on PyTorch (Paszke, 2019). All other details are deferred to the Appendix C and D.1.

## 5 EMPIRICAL STUDIES

### 5.1 EMPIRICAL ANALYSIS ON CIFAR

For a comprehensive comparison on CIFAR-10/100, we consider two categories of DP methods as baselines: static DP and dynamic DP. From static DP, we include 15 representative methods: static random pruning, CD (Agarwal et al., 2020), Herding (Welling, 2009), K-means (Sorscher et al., 2022), Least Confidence and Entropy (Coleman et al., 2019), Forgetting (Toneva et al., 2018), GraNd and EL2N (Paul et al., 2021), DeepFool (Ducoffe & Precioso, 2018), Craig (Mirzasoleiman et al., 2020), Glister (Killamsetty et al., 2021b), Influence (Koh & Liang, 2017), and DP (Yang et al., 2022). From dynamic DP, we adopt four methods: dynamic random pruning, $\epsilon$-greedy (Raju et al., 2021), UCB (Raju et al., 2021), and InfoBatch[3] (Qin et al., 2024), along with our proposed method **OrderDP**, which also belongs here.

**Performance comparison.** From Tables 1 and 2, our systematic study suggests the following trends: ① Dynamic random pruning outperforms static random by preserving higher sample diversity, and both $\epsilon$-greedy and UCB adaptively explore sample importance, but **OrderDP** consistently surpasses other baselines in accuracy and robustness across all prune ratios. ② At 30% pruning, only **OrderDP** matches full-data accuracy. ③ Under 50% and 70%, **OrderDP** has the smallest accuracy drop, outperforming both static and existing dynamic methods. ④ Compared to InfoBatch, **OrderDP** consistently yields higher accuracy as pruning becomes more aggressive.

**Efficiency comparison.** Table 3 reports end-to-end training time and GPU-hours under identical settings. **OrderDP** achieves the fastest training and lowest GPU-hours, improving upon InfoBatch without loss in accuracy. Additional CIFAR results and extended comparisons are in Appendix D.

**Extended comparison of varying pruning ratios.** To further evaluate the performance of **OrderDP**, we compare it with InfoBatch, a state-of-the-art data pruning algorithm, across different prune ratios on the CIFAR-10 and CIFAR-100 datasets. The results are demonstrated in Figure 3. It can be observed that **OrderDP** not only achieves higher accuracy at every pruning ratio, but also remains

---

[3]In the original experiments of InfoBatch (Qin et al., 2024), an annealing algorithm was incorporated. To ensure fair comparison, we have removed this component from all implementations.

Table 1: Static pruning results (accuracy, %) on `CIFAR10` and `CIFAR100` with ResNet-18. Accuracy (%, ↑). Best in **bold**. Performance gaps to full-data are in blue / orange.

| Dataset | CIFAR10 | | | CIFAR100 | | |
|---|---|---|---|---|---|---|
| Prune Ratio % | 30 | 50 | 70 | 30 | 50 | 70 |
| Static Random | $94.6_{\downarrow 1.0}$ | $93.3_{\downarrow 2.3}$ | $90.2_{\downarrow 5.4}$ | $73.8_{\downarrow 4.4}$ | $72.1_{\downarrow 6.1}$ | $69.7_{\downarrow 8.5}$ |
| CD (Agarwal et al., 2020) | $95.0_{\downarrow 0.6}$ | $94.3_{\downarrow 1.3}$ | $90.8_{\downarrow 4.8}$ | $74.2_{\downarrow 4.0}$ | $72.3_{\downarrow 5.9}$ | $70.3_{\downarrow 7.9}$ |
| Herding (Welling, 2009) | $92.2_{\downarrow 3.4}$ | $88.0_{\downarrow 7.6}$ | $80.1_{\downarrow 15.5}$ | $73.1_{\downarrow 5.1}$ | $71.8_{\downarrow 6.4}$ | $69.6_{\downarrow 8.0}$ |
| K-Center (Sener & Savarese, 2018) | $94.7_{\downarrow 0.9}$ | $93.9_{\downarrow 1.7}$ | $90.9_{\downarrow 4.7}$ | $74.1_{\downarrow 4.1}$ | $72.2_{\downarrow 6.0}$ | $70.2_{\downarrow 8.0}$ |
| Least Confidence (Coleman et al., 2019) | $95.0_{\downarrow 0.6}$ | $94.5_{\downarrow 1.1}$ | $90.3_{\downarrow 5.3}$ | $74.2_{\downarrow 4.0}$ | $72.3_{\downarrow 5.9}$ | $69.8_{\downarrow 8.4}$ |
| Margin (Coleman et al., 2019) | $94.9_{\downarrow 0.7}$ | $94.3_{\downarrow 1.3}$ | $90.9_{\downarrow 4.7}$ | $74.0_{\downarrow 4.2}$ | $72.2_{\downarrow 6.0}$ | $70.2_{\downarrow 8.0}$ |
| Forgetting (Toneva et al., 2018) | $94.7_{\downarrow 0.9}$ | $94.1_{\downarrow 1.5}$ | $91.7_{\downarrow 3.9}$ | $75.3_{\downarrow 2.9}$ | $73.1_{\downarrow 5.1}$ | $69.9_{\downarrow 8.3}$ |
| GraNd-4 (Paul et al., 2021) | $95.3_{\downarrow 0.3}$ | $94.6_{\downarrow 1.0}$ | $91.2_{\downarrow 4.4}$ | $74.6_{\downarrow 3.6}$ | $71.4_{\downarrow 6.8}$ | $68.8_{\downarrow 9.4}$ |
| DeepFool (Ducoffe & Precioso, 2018) | $95.1_{\downarrow 0.5}$ | $94.1_{\downarrow 1.5}$ | $90.0_{\downarrow 5.6}$ | $74.2_{\downarrow 4.0}$ | $73.2_{\downarrow 5.0}$ | $69.8_{\downarrow 6.4}$ |
| Craig (Mirzasoleiman et al., 2020) | $94.8_{\downarrow 0.8}$ | $93.3_{\downarrow 3.3}$ | $88.4_{\downarrow 7.2}$ | $74.4_{\downarrow 3.8}$ | $71.9_{\downarrow 6.3}$ | $69.7_{\downarrow 8.5}$ |
| Glister (Killamsetty et al., 2021b) | $95.2_{\downarrow 0.4}$ | $94.0_{\downarrow 1.6}$ | $90.9_{\downarrow 4.7}$ | $74.6_{\downarrow 3.6}$ | $73.2_{\downarrow 5.0}$ | $70.4_{\downarrow 7.8}$ |
| Influence (Koh & Liang, 2017) | $93.1_{\downarrow 2.5}$ | $91.3_{\downarrow 4.3}$ | $88.3_{\downarrow 7.3}$ | $74.4_{\downarrow 3.8}$ | $72.0_{\downarrow 6.2}$ | $68.9_{\downarrow 9.5}$ |
| EL2N-2 (Toneva et al., 2018) | $94.4_{\downarrow 1.2}$ | $93.2_{\downarrow 2.4}$ | $89.8_{\downarrow 5.8}$ | $74.1_{\downarrow 4.1}$ | $71.0_{\downarrow 7.2}$ | $68.5_{\downarrow 9.7}$ |
| EL2N-20 (Toneva et al., 2018) | $95.3_{\downarrow 0.3}$ | $95.1_{\downarrow 0.5}$ | $91.9_{\downarrow 3.7}$ | $77.2_{\downarrow 1.0}$ | $72.1_{\downarrow 6.1}$ | - |
| DP (Yang et al., 2023) | $94.9_{\downarrow 0.7}$ | $93.8_{\downarrow 1.8}$ | $90.8_{\downarrow 4.8}$ | $77.2_{\downarrow 1.0}$ | $73.1_{\downarrow 5.1}$ | - |
| OrderDP | $\mathbf{95.6}_{\uparrow 0.0}$ | $\mathbf{95.3}_{\downarrow 0.2}$ | $\mathbf{95.0}_{\downarrow 0.6}$ | $\mathbf{78.2}_{\uparrow 0.0}$ | $\mathbf{77.9}_{\downarrow 0.3}$ | $\mathbf{76.7}_{\downarrow 1.5}$ |
| Whole Dataset | $95.6_{\pm 0.1}$ | | | $78.2_{\pm 0.1}$ | | |

Table 2: Dynamic pruning results (accuracy, %) on `CIFAR10` and `CIFAR100` with ResNet-18 and ResNet-50. Accuracy (%, ↑). Best in **bold**. Performance gaps to full-data are in blue / orange.

| Dataset | CIFAR10 | | | | | | CIFAR100 | | | | | |
|---|---|---|---|---|---|---|---|---|---|---|---|---|
| Backbone | ResNet-18 | | | ResNet-50 | | | ResNet-18 | | | ResNet-50 | | |
| Prune Ratio % | 30 | 50 | 70 | 30 | 50 | 70 | 30 | 50 | 70 | 30 | 50 | 70 |
| Dynamic Random | $94.8_{\downarrow 0.8}$ | $94.5_{\downarrow 1.1}$ | $93.0_{\downarrow 2.6}$ | $95.1_{\downarrow 0.5}$ | $94.9_{\downarrow 0.7}$ | $93.6_{\downarrow 2.0}$ | $77.3_{\downarrow 0.9}$ | $75.3_{\downarrow 2.9}$ | $72.8_{\downarrow 5.4}$ | $77.9_{\downarrow 2.7}$ | $76.1_{\downarrow 4.5}$ | $73.9_{\downarrow 6.7}$ |
| $\epsilon$-greedy | $95.2_{\downarrow 0.4}$ | $94.9_{\downarrow 0.7}$ | $94.1_{\downarrow 1.5}$ | $95.4_{\downarrow 0.2}$ | $95.1_{\downarrow 0.5}$ | $94.3_{\downarrow 1.3}$ | $76.4_{\downarrow 1.8}$ | $74.8_{\downarrow 3.4}$ | $72.9_{\downarrow 5.3}$ | $77.2_{\downarrow 3.4}$ | $76.3_{\downarrow 4.3}$ | $74.1_{\downarrow 6.5}$ |
| UCB | $95.3_{\downarrow 0.3}$ | $94.7_{\downarrow 0.9}$ | $93.9_{\downarrow 1.7}$ | $95.5_{\downarrow 0.1}$ | $95.0_{\downarrow 0.6}$ | $94.2_{\downarrow 1.4}$ | $77.3_{\downarrow 0.9}$ | $75.3_{\downarrow 2.9}$ | $73.2_{\downarrow 5.0}$ | $78.0_{\downarrow 2.6}$ | $76.5_{\downarrow 4.1}$ | $74.3_{\downarrow 6.3}$ |
| InfoBatch | $95.6_{\downarrow 0.0}$ | $95.0_{\downarrow 0.6}$ | $94.5_{\downarrow 1.1}$ | $95.6_{\downarrow 0.0}$ | $95.3_{\downarrow 0.3}$ | $94.7_{\downarrow 0.9}$ | $78.1_{\downarrow 0.1}$ | $77.7_{\downarrow 0.5}$ | $75.9_{\downarrow 2.3}$ | $80.4_{\downarrow 0.2}$ | $78.6_{\downarrow 2.0}$ | $76.4_{\downarrow 4.2}$ |
| OrderDP | $\mathbf{95.6}_{\uparrow 0.0}$ | $\mathbf{95.3}_{\downarrow 0.2}$ | $\mathbf{95.0}_{\downarrow 0.6}$ | $\mathbf{95.6}_{\uparrow 0.0}$ | $\mathbf{95.4}_{\downarrow 0.2}$ | $\mathbf{95.0}_{\downarrow 0.6}$ | $\mathbf{78.2}_{\uparrow 0.0}$ | $\mathbf{77.9}_{\downarrow 0.3}$ | $\mathbf{76.7}_{\downarrow 1.5}$ | $\mathbf{80.6}_{\uparrow 0.0}$ | $\mathbf{79.8}_{\downarrow 0.8}$ | $\mathbf{77.9}_{\downarrow 2.7}$ |
| Whole Dataset | $95.6_{\pm 0.1}$ | | | $95.6_{\pm 0.1}$ | | | $78.2_{\pm 0.1}$ | | | $80.6_{\pm 0.1}$ | | |

comparable to other algorithms and reduces total training time in most cases. Moreover, InfoBatch cannot prune to an extreme ratio (limited by 77.9% on CIFAR-10 or 72.2% on CIFAR-100 in our setting) due to its fixed retention mechanism. *__OrderDP__ supports arbitrary pruning ratios because data retention is fully specified by the the exploration size s and exploitation size q (see Section 5.3).* These results confirm that **OrderDP** 's sample-selection strategy delivers near-optimal efficiency and robustness, making it particularly well-suited for resource-constrained scenarios where preserving accuracy is paramount.

## 5.2 EMPIRICAL ANALYSIS ON IMAGENET-1K

We evaluate Dynamic Random, UCB (Raju et al., 2021), InfoBatch (Qin et al., 2024) and our **OrderDP** on ImageNet-1K with ResNet-50 at 40% pruning (Table 3). **OrderDP** matches/exceeds all baselines in accuracy while not significantly increasing the total GPU runtime; it retains full-data performance at 40% pruning and incurs only a 0.4% drop at 60% (Table 4). ① *Efficiency:* **OrderDP** completes training faster and uses fewer GPU-hours than all competing methods. ② *Robustness:* It shows no loss at moderate prune ratios and only minimal degradation under aggressive pruning. Together, these findings confirm that **OrderDP** achieves near-lossless accuracy with a substantial reduction in compute, making it ideal for large-scale training under tight resource budgets.

## 5.3 ABLATION EXPERIMENT

We study how two-stage pruning decomposition, parameterized by the exploration size $s$ and exploitation size $q$, affects **OrderDP** 's performance on CIFAR-10/100 with ResNet-18 (Figure 4).

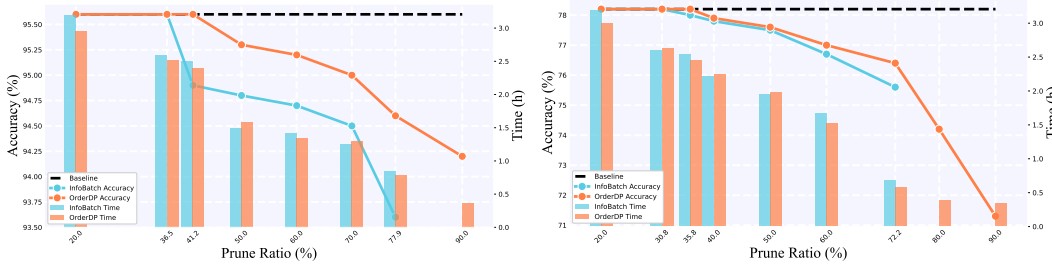

(a) Accuracy vs. Training Time on CIFAR-10      (b) Accuracy vs. Training Time on CIFAR-100

Figure 3: More accuracy and time results for different prune ratios on CIFAR-10/100 for **OrderDP** and InfoBatch, using ResNet-18. The lossless pruning ratios are marked in the figure.

Table 3: Comparison of performance and time cost on ImageNet-1K. Results are reported with ResNet-50 under 40% prune ratio for 90 epochs on a 2-L40-GPU server. "Total (n*h)" is the total node hour.

|  | Random | $\epsilon$-greedy | UCB | InfoBatch | Ours | Full Data |
|---|---|---|---|---|---|---|
| Acc (%) | $73.4_{\pm 0.3}$ | $75.2_{\pm 0.3}$ | $75.4_{\pm 0.3}$ | $75.6_{\pm 0.2}$ | $\mathbf{76.4}_{\pm 0.2}$ | $76.4_{\pm 0.2}$ |
| Time (h) | 21.1 | 21.1 | 21.1 | 21.6 | 21.5 | 35.2 |
| Total (n*h) | 42.2 | 42.2 | 42.2 | 43.2 | 43.0 | 70.4 |

Table 4: Experiments on ImageNet-1K. The models here are all implemented based on ResNet-50$_{\text{PyTorch}}$.

| Prune Ratio % | 30 | 40 | 60 |
|---|---|---|---|
| InfoBatch | $76.4_{\downarrow 0.0}$ | $75.6_{\downarrow 0.8}$ | $74.9_{\downarrow 1.5}$ |
| OrderDP | $\mathbf{76.4}_{\uparrow 0.0}$ | $\mathbf{76.4}_{\uparrow 0.0}$ | $76.0_{\downarrow 0.4}$ |
| Whole Dataset | | $76.4_{\pm 0.1}$ | |

**Fixed prune ratio** Under a fixed effective prune ratio, we vary the decomposition of the retained data portion by adjusting the exploration ratio $s/|D|$ and the exploitation ratio $q/s$, while keeping their product $(q/s) \cdot (s/|D|)$ unchanged.

As shown in Figure 4, **OrderDP** achieves identical accuracy across all decompositions on both datasets, demonstrating its precise control over the prune ratio. The training time remains stable across different decompositions, indicating consistent computational cost when the overall prune ratio is fixed.

**Varying prune ratios.** As the prune ratio grows, training time drops sharply while accuracy degrades more slowly—up to about 70% pruning, where we see over 95% on CIFAR-10 and over 76% on CIFAR-100 with half the compute. Beyond that, further pruning gives diminishing accuracy but continues to cut runtime. This shows **OrderDP** 's ability to trace a smooth efficiency–performance frontier and lets practitioners pick the "sweet spot" matching their compute budget.

Our ablation shows that decoupling exploration ($s$) and exploitation ($q$) achieves exact pruning ratios without efficiency loss and yields a smooth accuracy–cost frontier, enabling straightforward budget selection. We further provide stability results under multiple runs in Appendix D.6.

## 5.4 SENSITIVITY ANALYSIS

**Cross-architecture robustness evaluation.** Table 6 reports the maximum lossless prune ratios of InfoBatch and **OrderDP** on ResNet-18/50 across CIFAR-10, CIFAR-100, and ImageNet-1K. InfoBatch usually caps in the mid-30% range, while **OrderDP** extends this by 4–6 points, especially on harder datasets, showing its ability to prune more aggressively without accuracy loss.

We adopt the Timm (Wightman et al., 2021) ImageNet training stack, which combines mixed-precision training with strong augmentation and regularization methods such as MixUp, and CutMix (Zhong et al., 2017; Zhang et al., 2018; Yun et al., 2019), and observe that **OrderDP** continues to yield lossless speedups under this stronger recipe, indicating that it is compatible with existing acceleration and augmentation pipelines. Beyond CNN-based architectures, **OrderDP** also maintains lossless accuracy at

Table 5: Cross-architecture robustness evaluation on ImageNet-1K. ViT-Base (MAE) is pretrained with **OrderDP** for 300 epochs and fine-tuned for 100 epochs. Swin-Tiny is trained from scratch with **OrderDP**.

| Model | Prune Ratio | Original | OrderDP |
|---|---|---|---|
| R-50$_{\text{Timm}}$ | 29.8% | 78.4 | $78.3_{\downarrow 0.1}$ |
| Swin-T | 22.1% | 81.5 | $81.4_{\downarrow 0.1}$ |
| ViT-B (MAE) | 30.8% | 82.8 | $82.8_{\uparrow 0.0}$ |

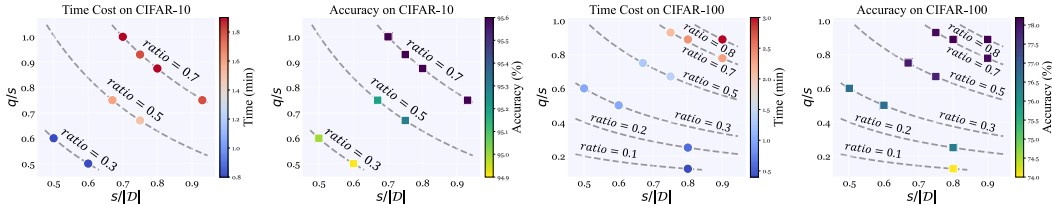

Figure 4: Performance with different ratio parameters. Here $(q/s) \cdot (s/|\mathcal{D}|)$ represents the retained data ratio, and thus the prune ratio is calculated as $1 - (q/s) \cdot (s/|\mathcal{D}|)$. Results are reported with ResNet-18.

Table 6: Cross-architecture robustness results of OrderDP. 'Full Dataset' denotes training on the original dataset without pruning.

|  | CIFAR-10 | | CIFAR-100 | | ImageNet-1K | |
|---|---|---|---|---|---|---|
|  | R-18 | R-50 | R-18 | R-50 | R-18 | R-50 |
| Full Dataset | 95.6 | 95.6 | 78.2 | 80.6 | 70.5 | 76.4 |
| InfoBatch | 95.5 | 95.6 | 78.2 | 80.6 | 70.4 | 76.4 |
| Saved (%) | 36.5 | 37.1 | 30.8 | 36.3 | 21.8 | 34.3 |
| **OrderDP** | 95.5 | 95.6 | 78.2 | 80.6 | 70.5 | 76.5 |
| Saved (%) | **41.2** | **42.6** | **35.8** | **41.1** | **27.3** | **39.8** |

Table 7: Comparison of accuracy (%) and saved cost (%) on CIFAR-10 when trained with R-50 using different optimizers.

|  | SGD | AdamW | LARS | LAMB |
|---|---|---|---|---|
| Full Dataset | 95.6 | 94.3 | 95.5 | 95.0 |
| InfoBatch | 95.6 | 94.3 | 95.5 | 95.0 |
| Saved (%) | 37.1 | 37.0 | 37.1 | 37.1 |
| OrderDP | 95.6 | 94.4 | 95.5 | 95.0 |
| Saved (%) | **42.6** | **42.4** | **42.5** | **42.6** |

**Note:** All the results are obtained from an 2-L40-GPU server.

20%–30% pruning on Swin-Tiny (Liu et al., 2021) and ViT-Base (MAE) (He et al., 2021) (Table 5), showing that the loss-based ordering remains stable on Vision Transformers, and that **OrderDP** naturally transfers across heterogeneous architectures as a plug-and-play module.

**Cross-optimizer robustness evaluation.** We test widely used optimizers—SGD (Bottou et al., 1991), AdamW (Loshchilov & Hutter, 2019), LARS (You et al., 2017), and LAMB (You et al., 2019)—on ResNet-50/CIFAR-10 (Table 7). InfoBatch saves 37.1% of training cost across all optimizers, while **OrderDP** raises savings to about 42.5%. This consistent gain shows that **OrderDP** 's dynamic sample selection is optimizer-agnostic: by focusing on high-loss examples, it reduces redundant computation and delivers plug-and-play efficiency without accuracy loss.

# 6 CONCLUSION

In this paper, we introduced **OrderDP**, a novel dynamic data pruning framework that provides rigorous theoretical guarantees while achieving substantial training acceleration. Unlike prior approaches, **OrderDP** ensures unbiased gradient estimation and offers exact control of the pruning ratio, leading to more stable and efficient data pruning. Our theoretical analysis establishes both convergence guarantees and generalization bounds, demonstrating its robustness across datasets and pruning levels. Empirically, **OrderDP** consistently attains equal or better accuracy than state-of-the-art baselines, while reducing runtime and overall computational cost by 40–45%. Moreover, its simpler plug-and-play design makes it easy to integrate into existing pipelines. These findings highlight the potential of our method as a scalable solution that balances efficiency, accuracy, and theoretical rigor.

## ETHICS STATEMENT

Our work focuses on improving training efficiency in deep learning through dynamic data pruning. All experiments are conducted on widely used public benchmark datasets (CIFAR-10, CIFAR-100, and ImageNet-1K), which do not involve any personally identifiable information, sensitive attributes, or human subjects. The study does not pose foreseeable risks related to privacy, fairness, or security. Moreover, no external sponsorship or conflicts of interest have influenced the design, analysis, or reporting of this work. As such, we believe our research complies with the ICLR Code of Ethics and raises no ethical concerns.

## REPRODUCIBILITY STATEMENT

We are committed to ensuring the reproducibility of our results. To this end, we provide detailed descriptions of datasets (CIFAR-10, CIFAR-100, ImageNet-1K), model architectures (ResNet-18, ResNet-50), hyperparameters, and training protocols in the main text and Appendix. For reproducibility, our implementation is based on PyTorch, with standard data augmentation (normalization, random cropping, horizontal flipping), SGD optimizer with momentum, weight decay, and OneCycle learning rate scheduling. We will submit the full source code and configuration files in the supplementary material to enable independent verification of our experiments. In addition, ablation studies and sensitivity analyses provide transparency into the robustness of our method across pruning ratios, optimizers, and architectures.

## ACKNOWLEDGMENTS

This work was supported in part by the BNBU Research Grant (No. of UICR0100031, UICR0700108-25, and UICR0900003) at Beijing Normal-Hong Kong Baptist University, Zhuhai, PR China. This research was also supported by the National Natural Science Foundation of China (Grant No. 12401415), the Natural Science Foundation of Hunan Province (Grant No. 2025JJ60009), and the Project of Scientific Research Fund of Hunan Provincial Science and Technology Department (Grant No. 25B0148).

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

# A    PROOF OF THEORETICAL ANALYSIS

## A.1    PROOF OF THEOREM 1

*Proof.* We just need to show that $\tilde{g}$ is an unbiased estimator of a sub-gradient of $L_q(\theta)$ at $\theta^t$, namely $E\tilde{g} \in \partial L_q(\theta^t)$. At first, it holds that

$$E\tilde{g}^t = \frac{1}{q} E \sum_{i \in Q} g_i^t = \frac{1}{q} \sum_{i=1}^n P(i \in Q) g_i^t$$

$$= \frac{1}{q} \sum_{j=1}^n P((j) \in Q) g_{(j)}^t ,$$

where $g_i^t \in \partial L_i(\theta^t)$ is a sub-gradient of $L_i$ at $\theta^t$. In the above equality chain, the third equality is simply the definition of expectation, and the last equality is because $((1), (2), \ldots, (n))$ is a permutation of $(1, 2, \ldots, n)$.

For any given index $j$, $P((j) \in Q) \neq \frac{q}{n}$ thus $E\tilde{g}^t \notin \partial L(\theta^t)$. To analyze $P((j) \in Q)$, define $A_j = ((1), (2), \ldots, (j-1))$, and $A_j^c = ((j+1), (j+2), \ldots, (n))$ then

$$
\begin{aligned}
P((j) \in Q) &= P\left((j) \in q\text{-argmax}_{i \in S} \mathcal{H}_i(\theta)\right) \\
&= P\left((j) \in S \text{ and } S \text{ contains at most } q - 1 \text{ items in } A_j\right) \\
&= P\left((j) \in S\right) P\left(S \text{ contains at most } q - 1 \text{ items in } A_j \mid (j) \in S\right) \\
&= P\left((j) \in S\right) \sum_{l=l_1}^{l_2} P\left((j) \text{ appears at } l \text{ position in } S \mid (j) \in S\right),
\end{aligned}
\tag{7}
$$

where $0 \leq l_1 \leq l_2 \leq s$ measures the possible positions of $(j) \in S$ These two variables vary depending on the choice of $(j)$. For examples, if $(j) = (1)$, $(j)$ should be included in $Q$ since $(1)$ would be at the top-1 position of $S$.

Notice that $S$ is randomly chosen from sample index set $(1, 2, \ldots, n)$ without replacement. There are in total $\binom{n}{s}$ different sets $S$ such that $|S| = s$. Among them, there are $\binom{n-1}{s-1}$ different sets $S$ which contains the index $(j)$, thus

$$P((j) \in S) = \frac{\binom{n-1}{s-1}}{\binom{n}{s}} .
\tag{8}$$

Given the condition $(j) \in S$, $(j)$ appears at $l$ position means $S$ contains $l - 1$ items in $A_j$ and $s - l$ items in $A_j^c$, thus we have the constraints:

$$s - l \leq n - j \quad \text{and} \quad 1 \leq l - 1 \leq j - 1.$$

Thus we conclude $s - n + j \leq l \leq j$, i.e., $l_1 = \max\{1, s - n + j\}$ and $l_2 = \min\{1, j\}$. There are $\binom{n-j}{s-l}$ such possible set $S$ for $(j) \in S$, whereby it holds that

$$P\left(S \text{ contains at most } q - 1 \text{ items in } A_j \mid (j) \in S\right)$$

$$= \sum_{l=l_1}^{l_2} P\left((j) \text{ appears at } l \text{ position in } S \mid (j) \in S\right)$$

$$= \sum_{l=\max\{1, s-n+j\}}^{\min\{q,j\}} \frac{\binom{j-1}{l-1}\binom{n-j}{s-l}}{\binom{n-1}{s-1}}
\tag{9}$$

Substituting Equations (7) and (8) into Equation (6), we arrive at

$$P((j) \in Q) = \frac{\binom{n-1}{s-1}}{\binom{n}{s}} \sum_{l=\max\{1, s-n+j\}}^{\min\{q,j\}} \frac{\binom{j-1}{l-1}\binom{n-j}{s-l}}{\binom{n-1}{s-1}} = \sum_{l=\max\{1, s-n+j\}}^{\min\{q,j\}} \frac{\binom{j-1}{l-1}\binom{n-j}{s-l}}{\binom{n}{s}} = \gamma_j.
\tag{10}$$

Therefore,

$$E\tilde{g}^t = \frac{1}{q} \sum_{j=1}^n P((j) \in Q) g_{(j)}^t = \frac{1}{q} \sum_{j=1}^n \gamma_j g_{(j)}^t \in \partial L_q(\theta^t) ,
\tag{11}$$

where the last inequality is due to the additivity of sub-gradient (for both convex and weakly convex function) $\square$.   $\square$

## A.2 PROOF OF PROPOSITION 2

*Proof.* We will show that

$$\lim_{j,n\to\infty,j/n=z} \gamma_j = \frac{1}{n}\frac{s!}{(l-1)!(s-l)!}\sum_{l=1}^{q}\left(\frac{j}{n}\right)^{l-1}\left(1-\frac{j}{n}\right)^{s-l}. \tag{12}$$

We begin the proof by changing the variable $z = \frac{j}{n}$.

At first, the Stirling's approximation yields that when $n$ and $j$ are both sufficiently large, it holds that

$$\binom{n}{j} \sim \sqrt{\frac{n}{2\pi j(n-j)}} \frac{n^n}{j^j(n-j)^{n-j}}. \tag{13}$$

Thus,

$$\lim_{j,n\to\infty,j/n=z} \frac{\binom{n-s}{j-l}}{\binom{n-1}{j-1}} = \frac{\frac{n^{n-s}}{j^{j-l}(n-j)^{n-j-s+l}}}{\frac{n^{n-1}}{j^{j-1}(n-j)^{n-j}}} = \frac{j^{l-1}(n-j)^{s-l}}{n^{s-1}} = \left(\frac{j}{n}\right)^{l-1}\left(\frac{n-j}{n}\right)^{s-l} \tag{14}$$

where the first equality utilizes Equation (10) and the fact that $s, l, 1$ are negligible in the limit case (except the exponent terms).

On the other hand, it holds by rearranging the factorial numbers that

$$\frac{1}{n}\frac{\binom{n-s}{j-l}}{\binom{n-1}{j-1}}\frac{s!}{(l-1)!(s-l)!} = \frac{\binom{j-1}{l-1}\binom{n-j}{s-l}}{\binom{n}{s}}. \tag{12}$$

Recall $\gamma_j = \sum_{l=\max\{1,s-n+j\}}^{\min\{q,j\}}\frac{\binom{j-1}{l-1}\binom{n-j}{s-l}}{\binom{n}{s}}$. Let

$$\gamma_j = \sum_{l=\max\{1,s-n+j\}}^{\min\{q,j\}}\frac{\binom{j-1}{l-1}\binom{n-j}{s-l}}{\binom{n}{s}} = \sum_{l=1}^{q}\frac{\binom{j-1}{l-1}\binom{n-j}{s-l}}{\binom{n}{s}}, \tag{15}$$

where we set the value to 0 for $l \in [1, s-n+j]$ and $[j, q]$ if $s-n+j > 1$ and $j < q$. Therefore, we conclude the following by noticing $s > q$,

$$\begin{aligned}
\frac{d}{dz}\gamma(z) &= \sum_{l=2}^{q}(l-1)z^{l-2}(1-z)^{s-l}\frac{s!}{(l-1)!(s-l)!} - \sum_{l=1}^{q}(s-l)z^{l-1}(1-z)^{s-l-1}\frac{s!}{(l-1)!(s-l)!}\\
&= \sum_{l=2}^{q}z^{l-2}(1-z)^{s-l}\frac{s!}{(l-2)!(s-l)!} - \sum_{l=1}^{q}z^{l-1}(1-z)^{s-l-1}\frac{s!}{(l-1)!(s-l-1)!}\\
&= \sum_{l=1}^{q-1}z^{l-1}(1-z)^{s-l-1}\frac{s!}{(l-1)!(s-l-1)!} - \sum_{l=1}^{q}z^{l-1}(1-z)^{s-l-1}\frac{s!}{(l-1)!(s-l-1)!}\\
&= -z^{q-1}(1-z)^{s-q-1}\frac{s!}{(q-1)!(s-q-1)!}\\
&= -z^{q-1}(1-z)^{s-q-1}\frac{(s-1)!}{(q-1)!(s-q-1)!}s.
\end{aligned} \tag{16}$$

In other words, $1 - \frac{1}{s}\gamma(z)$ is the cumulative distribution function of $\text{Beta}(q, s-q)$ when $n \to \infty$. $\square$

## A.3 PROOF OF THEOREM 3

Full version of Theorem 3: Let $(\theta^t)_{t=0}^{T}$ be the sequence generated by Algorithm 1. Suppose there exists a finite $\theta^* \in \arg\min_\theta \mathcal{L}_q(\theta)$, $\mathcal{L}_q(\theta^*) < \infty$. If each $\mathcal{L}_i(\cdot)$ is convex and $G$-Lipschitz, then

$$\min_{0\le t\le T}\mathbb{E}\left[\mathcal{L}_q(\theta^t) - \mathcal{L}_q(\theta^*)\right] \le \frac{\eta_{\max}\left(\|\theta^1 - \theta^*\|_2^2 + G^2\sum_{t=1}^{T}(\eta^t)^2\right)}{2\,\eta_{\min}\sum_{t=1}^{T}\eta^t}. \tag{17}$$

Moreover, if we define the weighted average $\bar{\theta}^T = \frac{1}{\sum_{t=1}^T \eta^t} \sum_{t=1}^T \eta^t \theta^t$, then

$$\mathbb{E}\left[\mathcal{L}_q(\bar{\theta}^T) - \mathcal{L}_q(\theta^*)\right] \leq \frac{\|\theta^1 - \theta^*\|_2^2 + G^2 \sum_{t=1}^T (\eta^t)^2}{2 \sum_{t=1}^T \eta^t}. \tag{18}$$

*Proof.* Consider the one update at epoch $t$, we have

$$\|\theta^{t+1} - \theta^*\|_2^2 = \|\theta^t - \theta^*\|_2^2 - 2\eta^t \langle \tilde{g}^t, \theta^t - \theta^* \rangle + (\eta^t)^2 \|\tilde{g}^t\|_2^2. \tag{19}$$

Taking the conditional expectation of $v^t$ given $\theta$ of equation 19 yields

$$\mathbb{E}[\|\theta^{t+1} - \theta^*\|_2^2] \leq \|\theta^t - \theta^*\|_2^2 - 2\eta^t \langle \mathbb{E}[\tilde{g}^t], \theta^t - \theta^* \rangle + (\eta^t)^2 G^2, \tag{20}$$

where we use $\|\tilde{g}^t\|_2 \leq G$. Because we maintain an unbiased gradient estimator $\mathbb{E}[\tilde{g}^t] \in \partial \mathcal{L}_q(\theta^t)$, we have that with convexity of $\mathcal{L}_q$, we have

$$-\langle \mathbb{E}[\tilde{g}^t], \theta^t - \theta^* \rangle \leq -(\mathcal{L}_q(\theta^t) - \mathcal{L}_q(\theta^*)), \tag{21}$$

where the $\theta^* \overset{\text{def}}{:=} \arg\min_\theta \mathcal{L}_q(\theta)$. Substituted equation 21 into equation 20 gives

$$\mathbb{E}\left[\|\theta^{t+1} - \theta^*\|_2^2\right] \leq \|\theta^t - \theta^*\|_2^2 - 2\eta^t \left(\mathcal{L}_q(\theta^t) - \mathcal{L}_q(\theta^*)\right) + (\eta^t)^2 G^2.$$
$$\implies 2\eta^t (\mathcal{L}_q(\theta^t) - \mathcal{L}_q(\theta^*)) \leq \left(\|\theta^t - \theta^*\|_2^2 - \mathbb{E}[\|\theta^{t+1} - \theta^*\|_2^2]\right) + (\eta^t)^2 G^2. \tag{22}$$

Take the expectation over the entire sequence $\theta^1, \ldots, \theta^{t+1}$, sum over $t = 1, \ldots, T$, we have

$$2\sum_{t=1}^T \eta^t \mathbb{E}[\mathcal{L}_q(\theta^t) - \mathcal{L}_q(\theta^*)] \leq \left(\|\theta^1 - \theta^*\|_2^2 - \mathbb{E}[\|\theta^{T+1} - \theta^*\|_2^2]\right) + G^2 \sum_{t=1}^T (\eta^t)^2. \tag{23}$$

It shows that

$$\frac{1}{\sum_{t=1}^T \eta^t} \sum_{t=1}^T \eta^t \mathbb{E}[\mathcal{L}_q(\theta^t) - \mathcal{L}_q(\theta^*)] \leq \frac{\|\theta^1 - \theta^*\|_2^2 + G^2 \sum_{t=1}^T (\eta^t)^2}{2 \sum_{t=1}^T \eta^t}. \tag{24}$$

With the observation that

$$\frac{1}{\sum_{t=1}^T \eta^t} \sum_{t=1}^T \eta^t \mathbb{E}[\mathcal{L}_q(\theta^t) - \mathcal{L}_q(\theta^*)]$$
$$= \frac{1}{\frac{1}{T}\sum_{t=1}^T \eta^t} \frac{1}{T} \sum_{t=1}^T \eta^t \mathbb{E}[\mathcal{L}_q(\theta^t) - \mathcal{L}_q(\theta^*)]$$
$$\geq \frac{\min_{1 \leq t \leq T} \eta^t \mathbb{E}[\mathcal{L}_q(\theta^t) - \mathcal{L}_q(\theta^*)]}{\max_{1 \leq t \leq T} \eta^t}$$
$$\geq \frac{\eta_{min}}{\eta_{max}} \min_{1 \leq t \leq T} \mathbb{E}[\mathcal{L}_q(\theta^t) - \mathcal{L}_q(\theta^*)], \tag{25}$$

where $\eta_{min} = \min_{1 \leq t \leq T} \eta^t$ and $\eta_{max} = \max_{1 \leq t \leq T} \eta^t$. Therefore, it holds that

$$\frac{\eta_{min}}{\eta_{max}} \min_{1 \leq t \leq T} \mathbb{E}[\mathcal{L}_q(\theta^t) - \mathcal{L}_q(\theta^*)] \leq \frac{1}{\sum_{t=1}^T \eta^t} \sum_{t=1}^T \eta^t \mathbb{E}[\mathcal{L}_q(\theta^t) - \mathcal{L}_q(\theta^*)]$$
$$\leq \frac{\|\theta^1 - \theta^*\|_2^2 + G^2 \sum_{t=1}^T (\eta^t)^2}{\sum_{t=1}^T \eta^t}. \tag{26}$$

Then we can derive that

$$\min_{1 \leq t \leq T} \mathbb{E}[\mathcal{L}_q(\theta^t) - \mathcal{L}_q(\theta^*)] \leq \frac{\eta_{max}(\|\theta^1 - \theta^*\|_2^2 + G^2 \sum_{t=1}^T (\eta^t)^2)}{2\eta_{min} \sum_{t=1}^T \eta^t}. \tag{27}$$

$\square$

## A.4 PROOF OF THEOREM 4

*Proof.* We begin this proof by leveraging the concepts of spectral risk measure (Acerbi & Tasche, 2002; Mehta et al., 2023). the surrogate loss $\mathcal{L}_q(\theta, D) = \sum_{i=1}^n \frac{\gamma_i}{q} L_{(i)}(\theta) = \sum_{i=1}^n \sigma_j Z_{(i)}$ is called an $L$-risk with a spectrum $\sigma_i = \frac{\gamma_i}{q}$ and $Z_{(i)} = L_{(i)}(\theta)$ for $i \in [n]$, which can be regarded as a functional of the CDF known as a *spectral risk measure*. $\{Z_i\}_{i=1}^n$ are arbitrary real-valued i.i.d. random variables drawn from CDF $F$. For our case, these refer to data instance $D_i$ of $n$ samples drawn from distribution $\mathcal{D}$ under parameter vector $\theta$.

Let $F_n(z) := \frac{1}{n} \sum_{i=1}^n 1_{(-\infty, z]}(Z_i)$ denote the (random) empirical CDF of the sample and define the empirical *quantile function* (or inverse CDF) as

$$F_n^{-1}(t) := \inf\{z : F_n(z) \geq t\} \quad \text{for } t \in (0, 1). \tag{28}$$

The population quantile function is defined similarly as

$$F^{-1}(t) := \inf\{z : F(z) \geq t\}. \tag{29}$$

The empirical quantile function can be written in terms of the order statistics as $F_n^{-1}(t) = Z_{(\lceil nt \rceil)}$. Notice in particular that when $t \in \left(\frac{i-1}{n}, \frac{i}{n}\right)$, we have that $F_n^{-1}(t) = Z_{(i)}$, where end-points are chosen to make $F_n^{-1}$ left continuous.

The spectrum $\sigma$ of an $L$-risk is typically defined as a discretization of a probability density $s$ on $(0, 1)$, such that

$$\sigma_i = \int_{(i-1)/n}^{i/n} s(t)\, dt, \tag{30}$$

so that it need not be redefined for every $n$. Given both the construction of $s$ and $F_n^{-1}$, we can rewrite the $L$-risk as

$$\begin{aligned}
\mathcal{L}_q(\theta, D) = \sum_{i=1}^n \sigma_i Z_{(i)} &= \sum_{i=1}^n \left( \int_{(i-1)/n}^{i/n} s(t)\, dt \right) Z_{(i)} \\
&= \sum_{i=1}^n \left( \int_{(i-1)/n}^{i/n} s(t) \cdot Z_{(\lceil nt \rceil)}\, dt \right) \\
&= \int_0^1 s(t) \cdot F_n^{-1}(t)\, dt =: \mathbb{L}_s[F_n],
\end{aligned} \tag{31}$$

where $\mathbb{L}_s[G] := \int_0^1 s(t) G^{-1}(t)\, dt$ is called a spectral risk measure with spectrum $s$ applied to CDF $G$.

It stands to reason that $\mathbb{L}_s[F_n]$ converges to $\mathbb{L}_s[F]$ in an appropriate sense. This convergence is governed by the Wasserstein distance between the empirical and population distribution, which we briefly recall here. In this section, we control the bias term appearing in the convergence analysis. The following lemmas consider a set of real numbers, representing losses at a single $\theta \in \mathbb{R}^d$. Let $x_1, \ldots, x_n \in \mathbb{R}$ be call the *full batch*, and let $X_1, \ldots X_m$ be a random sample selected uniformly *without* replacement from $\{x_1, \ldots, x_n\}$, called the *minibatch*. Let

$$F_n(x) := \frac{1}{n} \sum_{i=1}^n 1_{(-\infty, x]} \text{ and } F_{n,m}(x) := \frac{1}{m} \sum_{j=1}^m 1_{(-\infty, x)} \tag{32}$$

be the empirical CDFs, and let

$$F_n^{-1}(t) := \inf\{x : F_n(x) \geq t\} \text{ and } F_{n,m}(t) := \inf\{x : F_{n,m}(x) \geq t\}. \tag{33}$$

be the empirical quantile functions of the full batch and minibatch, respectively. Similarly, let

$$\mu_n := \sum_{i=1}^n \delta_{x_i} \text{ and } \mu_{n,m} = \sum_{j=1}^m \delta_{X_j} \tag{34}$$

be the empirical measures of the full batch and minibatch, respectively, with $\delta_x$ indicating a Dirac point mass at $x$. Let $u(t) := 1_{(0,1)} t$ be the uniform spectrum.

Recall the expressions of the $\mathcal{L}$-risk. We denote $\mathcal{L}(\theta^*) = \mathbb{E}_{D \sim \mathcal{D}}[\mathcal{L}(\theta^*, D)]$ as the optimal value. For the sampled distribution, we have

$$\mathbb{E}[\mathcal{L}_q(\theta^t, D)] = \mathbb{E}\left[\sum_{j=1}^{s} \frac{\hat{\gamma}_j}{q} \mathcal{L}_{i_{(j)}}(\theta^t)\right] = \mathbb{E}[\mathbb{L}_s[F_{n,s}(\cdot; \theta^t)]], \tag{35}$$

and for the uniform distribution, we define

$$\mathbb{E}[\mathcal{L}_u(\theta^t, D)] = \mathbb{E}\left[\sum_{j=1}^{s} \frac{1}{s} \mathcal{L}_{i_{(j)}}(\theta^t)\right] = \mathbb{E}[\mathbb{L}_u[F_{n,s}(\cdot; \theta^t)]]. \tag{36}$$

Moreover, the full-batch loss satisfies

$$\mathcal{L}(\theta^t, D) = \mathcal{L}_u(\theta^t, D) = \mathbb{L}_u[F_n(\cdot; \theta^t)]. \tag{37}$$

Here, the distributions $s$ and $u$ correspond to the sampling distribution of $\gamma$ in $\mathcal{L}_q(\theta^t, D)$ at step $t$ and the uniform distribution, respectively. The expectation is taken over the minibatch $\{i_1, \ldots, i_s\}$.

Therefore, we establish the generalization error over the set $\Theta := \{\theta^i\}_{i=1}^{T}$.

$$
\begin{aligned}
&\mathcal{L}(\theta^*) - \mathbb{E}[\mathcal{L}_q(\theta^t, D)] \\
&\leq \sup_{\theta \in \Theta} \mathcal{L}(\theta^*) - \mathbb{E}[\mathcal{L}_q(\theta^t, D)] \\
&= \sup_{\theta \in \Theta} \mathcal{L}(\theta^*) - \mathbb{E}[\mathbb{L}_s[F_{n,s}]] - \mathbb{L}_s[F_n] + \mathbb{L}_s[F_n] - \mathbb{E}[\mathbb{L}_u[F_{n,s}]] + \mathbb{E}[\mathbb{L}_u[F_{n,s}]] - \mathbb{L}_u[F_n] + \mathbb{L}_u[F_n] \\
&= \sup_{\theta \in \Theta} \mathcal{L}(\theta^*) - \mathbb{E}[\mathbb{L}_u[F_{n,s}]] + \mathbb{L}_u[F_n] - \mathbb{L}_s[F_n] \\
&\quad - \left(\mathbb{E}[\mathbb{L}_s[F_{n,s}]] - \mathbb{L}_s[F_n] - \left(\mathbb{E}[\mathbb{L}_u[F_{n,s}]] - \mathbb{L}_u[F_n]\right)\right) \\
&\leq \sup_{\theta \in \Theta} \mathcal{L}(\theta^*) - \mathbb{E}[\mathbb{L}_u[F_{n,s}]] + \sup_{\theta \in \Theta} \left(\mathbb{L}_u[F_n] - \mathbb{L}_s[F_n]\right) \\
&\quad + \sup_{\theta \in \Theta} \left(-\mathbb{E}[\mathbb{L}_s[F_{n,s}]] + \mathbb{L}_s[F_n] + \mathbb{E}[\mathbb{L}_u[F_{n,s}]] - \mathbb{L}_u[F_n]\right) \\
&\leq \inf_{\theta \in \Theta} \left|\mathbb{E}[\mathbb{L}_u[F_{n,s}]] - \mathcal{L}(\theta^*)\right| + \sup_{\theta \in \Theta} \left(\mathbb{L}_u[F_n] - \mathbb{L}_s[F_n]\right) \\
&\quad + \sup_{\theta \in \Theta} \left|\mathbb{E}[\mathbb{L}_s[F_{n,s}]] - \mathbb{L}_s[F_n] - \left(\mathbb{E}[\mathbb{L}_u[F_{n,s}]] - \mathbb{L}_u[F_n]\right)\right| \\
&\leq \frac{\eta_{\max}\left(\|\theta^1 - \theta^*\|_2^2 + G^2 \sum_{t=1}^{T}(\eta^t)^2\right)}{2\eta_{\min} \sum_{t=1}^{T} \eta^t} \\
&\quad + \sup_{\theta \in \Theta} \left(\mathbb{L}_u[F_n] - \mathbb{L}_s[F_n]\right) + \sup_{\theta \in \Theta} \|s - u\|_\infty \mathbb{E}[\|F_{n,m}^{-1} - F_n^{-1}\|_1] \\
&\leq \underbrace{\frac{\eta_{max}(\|\theta^1 - \theta^*\|_2^2 + G^2 \sum_{t=1}^{T}(\eta^t)^2)}{2\eta_{min} \sum_{t=1}^{T} \eta^t}}_{\text{unbiased part}} \underbrace{- \mathcal{Q}_n(\theta^t; s, q) + \sqrt{2} C_s B \sqrt{\frac{n-s}{s(n-1)}}}_{\text{biased part}},
\end{aligned}
$$

$$\tag{38}$$

where the third inequality follows the Theorem 3 and the fourth inequality follows Lemma 14 in (Mehta et al., 2023). We denote $C_s = \sup_{t \in (0,1)} |s(t) - u(t)|$, $B = \inf_{\theta \in [1,T]} \max_{i \in [1,n]} |\mathcal{L}_i(\theta, z_i)| < \infty$, and $\mathcal{Q}_n(\theta; s, q) := \inf_{\theta \in \Theta} \sum_{i=1}^{n} \left(\frac{r_i(\theta, D)}{q} - \frac{1}{n}\right) \mathcal{L}_i(\theta, z)$.

Moreover, if we use the weighted average $\bar{\theta}^T = \frac{1}{\sum_{t=1}^{T} \eta^t} \sum_{t=1}^{T} \eta^t \theta^t$ as output of **OrderDP**, the dependence on $\eta_{\max}$ and $\eta_{\min}$ can be removed and it holds that:

$$
\begin{aligned}
&\mathcal{L}(\theta^*) - \mathbb{E}[\mathcal{L}_q(\bar{\theta}^T, D)] \\
=&\mathcal{L}(\theta^*) - \mathbb{E}[\mathbb{L}_s[F_{n,s}]] - \mathbb{L}_s[F_n] + \mathbb{L}_s[F_n] - \mathbb{E}[\mathbb{L}_u[F_{n,s}]] + \mathbb{E}[\mathbb{L}_u[F_{n,s}]] - \mathbb{L}_u[F_n] + \mathbb{L}_u[F_n] \\
=&\mathcal{L}(\theta^*) - \mathbb{E}[\mathbb{L}_u[F_{n,s}]] + \mathbb{L}_u[F_n] - \mathbb{L}_s[F_n] - \left[\mathbb{E}[\mathbb{L}_s[F_{n,s}]] - \mathbb{L}_s[F_n] - (\mathbb{E}[\mathbb{L}_u[F_{n,s}]] - \mathbb{L}_u[F_n])\right] \\
\leq& \left|\mathbb{E}[\mathbb{L}_u[F_{n,s}]] - \mathcal{L}(\theta^*)\right| + \left(\mathbb{L}_u[F_n] - \mathbb{L}_s[F_n]\right) \\
& + \left|\mathbb{E}[\mathbb{L}_s[F_{n,s}]] - \mathbb{L}_s[F_n] - (\mathbb{E}[\mathbb{L}_u[F_{n,s}]] - \mathbb{L}_u[F_n])\right| \\
\leq& \left|\mathbb{E}[\mathbb{L}_u[F_{n,s}]] - \mathcal{L}(\theta^*)\right| + \sup_{\theta \in \Theta}\left(\mathbb{L}_u[F_n] - \mathbb{L}_s[F_n]\right) \\
& + \left|\mathbb{E}[\mathbb{L}_s[F_{n,s}]] - \mathbb{L}_s[F_n] - (\mathbb{E}[\mathbb{L}_u[F_{n,s}]] - \mathbb{L}_u[F_n])\right| \\
\leq& \underbrace{\frac{(\|\theta^1 - \theta^*\|_2^2 + G^2 \sum_{t=1}^T (\eta^t)^2)}{2\sum_{t=1}^T \eta^t}}_{\text{unbiased part}} \underbrace{- \mathcal{Q}_n(\theta^t; s, q) + \sqrt{2}C_s B \sqrt{\frac{n-s}{s(n-1)}}}_{\text{biased part}}.
\end{aligned}
$$

(39)

where the last inequality follows from (18) and other terms remain unchanged.

$\square$

## B  RELATED WORKS

**Static Data Pruning.** Static pruning techniques aim to pre-select a compact subset of the training data that can approximate the utility of the full dataset. A wide range of criteria have been proposed for this purpose. Diversity-based methods such as Contextual Diversity (CD) (Agarwal et al., 2020), Herding (Welling, 2009), and k-Center (Sener & Savarese, 2018) remove redundant samples by ensuring broad feature-space coverage. Difficulty-based strategies including Cal (Margatina et al., 2021) and Deepfool (Ducoffe & Precioso, 2018) prioritize hard-to-learn examples near decision boundaries. Error- and gradient-driven approaches such as GraNd and EL2N (Paul et al., 2021) and MOSO (Tan et al., 2023) instead exploit training dynamics or loss sensitivity. In parallel, uncertainty-based sampling (Coleman et al., 2019), influence-function analysis (Koh & Liang, 2017), and gradient matching approaches like GradMatch (Killamsetty et al., 2021b;a) provide alternative means of quantifying informativeness. More principled frameworks include bilevel optimization (Killamsetty et al., 2021b) and submodular subset selection (Iyer et al., 2021), where algorithms such as FL and Graph Cut (GC) (Iyer et al., 2021) explicitly balance coverage and information gain. Early computer vision work such as (Huh et al., 2016) also emphasized the importance of dataset diversity for transferable representations. While effective in specific cases, static approaches often require costly pre-computation, and their heuristics may not generalize well across architectures or datasets, particularly at ImageNet scale.

**Dynamic Data Pruning.** Dynamic methods instead make pruning decisions adaptively during training by leveraging information from the evolving model state. Early efforts such as ActiveBias (Chang et al., 2017) adjusted sampling probabilities based on prediction confidence, while forgetting-based measures (Toneva et al., 2018) revealed that unstable or frequently forgotten examples often provide valuable signal. Raju et al. (Raju et al., 2021) introduced exploration-based policies such as $\epsilon$-greedy and UCB, where uncertainty estimates guide the retention of high-value samples. Recent work has also examined improving random sampling policies themselves: Okanovic et al. (Okanovic et al., 2024) showed that repeated random sampling can significantly reduce time-to-accuracy, offering a complementary perspective to loss-based dynamic pruning approaches such as InfoBatch and **OrderDP**. More recently, InfoBatch (Qin et al., 2024) proposed an unbiased gradient estimator, showing that loss-based pruning can accelerate training without compromising accuracy on benchmarks like CIFAR-10/100 and ImageNet-1K. Building on this line of research, Yang et al. (Yang et al., 2022) and Sorscher et al. (Sorscher et al., 2022) extended dynamic pruning principles to large-scale pretraining, while He et al. (He et al., 2024) incorporated dynamically updated uncertainty estimates. In particular, Sorscher et al. (Sorscher et al., 2022) demonstrated that the optimal choice between hard and easy samples can depend on dataset scale, an observation that is complementary to the top-$q$ strategy analyzed in this work. Despite these advances, dynamic methods still face challenges: the

achievable "lossless" pruning ratio on new datasets is unpredictable, sorting operations can become expensive at scale, and empirical instability often emerges under aggressive pruning ratios. Ayed and Hayou (Ayed & Hayou, 2023) further analyze the fundamental bias of score-based pruning and show that reweighting can recover unbiasedness with respect to the original loss. Their perspective is complementary to ours: while they study limitations under $L$, we provide guarantees for a ranking-induced surrogate $L_q$ whose gap to $L$ is explicitly controlled.

**Cross-domain Data Selection and Pruning.** Beyond computer vision, pruning and selection strategies have been expanded to other domains such as NLP and speech. In speech recognition, unsupervised data selection has been explored through discrete speech units (Lu et al., 2022). For large-scale NLP pretraining, several studies investigate pruning and mixture optimization to accelerate convergence. (Marion et al., 2023) explored pruning strategies for pretraining corpora, while (Xie et al., 2023) introduced DoReMi, a framework that dynamically optimizes data mixtures for faster language model pretraining. Instruction tuning has further motivated task-specific pruning, exemplified by (Cao et al., 2023), who proposed instruction mining to select relevant subsets for downstream tasks. These works highlight that pruning is not limited to vision but constitutes a broader principle of efficient data utilization across modalities.

## C  EXPERIMENTAL INFRASTRUCTURES

**Software infrastructures.** All experiments are implemented in Python 3.12.4 using PyTorch 2.3.1 with CUDA 11.8 support. Key libraries include NumPy 1.26.4, pandas 2.2.3, torchvision 0.18.1, matplotlib 3.10.1, and scikit-learn 1.6.1 for data processing and analysis. We also employ `accelerate` 1.6.0 for multi-GPU training and `tqdm` 4.67.1 for progress visualization.

**Hardware infrastructures.** We conduct all experiments on a computer server with 2 NVIDIA L40 GPUs (with 48GB memory each), a single Intel Xeon Gold 6448Y CPU (32 physical cores), and 944 GiB of system RAM.

## D  ADDITIONAL EMPIRICAL RESULTS

### D.1  EXPERIMENTAL SETUP DETAILS

We provide software/hardware infrastructures in Appendix C; here we detail dataset-specific training setups throughout this paper.

**CIFAR-10/100:** The CIFAR-10/100 experiment with ResNet-18 can be reproduced with SGD using a maximum learning rate of 0.2 for the OneCycle scheduler under a batch size of 128. For experiments with ResNet-50, SGD is used with a maximum learning rate of 0.03 and batch size of 128 for baseline, InfoBatch, and **OrderDP**.

**ImageNet-1K:** The tests are implemented based on `Pytorch/examples`. The LARS optimizer and a maximum learning rate of 6.4 / 1.98 are used for batch size 1024 on ImageNet-1K experiments with ResNet-50/18.

### D.2  VALIDATION OF THEORETICAL PROPERTIES

Both Figure 5 and Figure 6 illustrate theoretical properties derived in Appendix A. Figure 5 empirically validates Proposition 2 by fixing $(s, q) = (100, 30)$ and increasing $n$, showing how $n\,\gamma_j$ converges to $\gamma(z)$ as $n$, $s$, and $q$ increase. Figure 6 illustrates Theorem 4 by comparing $\gamma_j$ to uniform sampling for different $q$ (fix $(n, s) = (200, 100)$ and increase $q \to 100$), highlighting that the gap between the two distributions vanishes (i.e., $\mathcal{Q}_n(\theta; s, q)$ and $C_s$ approach 0) as $q$ increases.

In addition to the above validations, we further examine the normalization behavior of the weights $\{\gamma_j\}$ derived in Eq. (10). Although providing a fully symbolic proof for the combinatorial form of $\gamma_j$ is algebraically involved, the construction of Algorithm 1 implies that exactly $q$ samples are selected at each iteration, suggesting that $\sum_{j=1}^{n} \gamma_j$ should be close to $q$, and therefore $\sum_{j=1}^{n} \gamma_j / q \approx 1$.

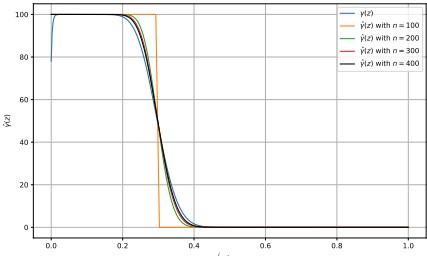 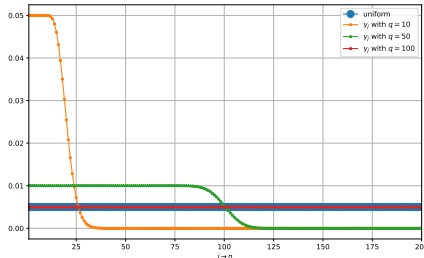

Figure 5: Empirical weight decay curves $n\,\gamma_j$ versus normalized index $j/n$, demonstrating convergence to the limiting density $\gamma(z)$ and the smoothing of the "cliff" as $n$ increases.

Figure 6: Comparison of the sampling weights $\gamma_j$ against uniform sampling for different exploitation sizes $q$, illustrating the deviation captured by the bias term $C_s$ in Theorem 4.

Figure 7 plots the normalized distributions $\gamma_j/q$ for different values of $q$ (with $(n, s) = (400, 100)$). As $q$ increases, the curves gradually flatten and approach the uniform distribution, consistent with Theorem 4.

Figure 8 further shows the empirical values of $\sum_{j=1}^{n} \gamma_j/q$ across $q \in \{10, 20, \ldots, 100\}$, all of which lie extremely close to 1 (within floating-point error). This provides strong numerical evidence that the weights induced by Algorithm 1 are properly normalized in practice.

*Proof.* We also provide proof of the claim $\sum_{j=1}^{n} \gamma_j/q = 1$ via Mathematical Induction.

For any $n, s$, when $q = 1$, we can show exactly that

$$\sum_j \frac{\gamma_j}{q} = \sum_j \frac{\binom{n-1}{s-1}}{\binom{n}{s}} = 1.$$

Suppose $\sum_j \gamma_j/q = 1$ for any $q = m$ where $m \in \mathbb{N}$ and $1 \le m \le s - 1$, we show $\sum_j \gamma_j/q = 1$ for $q = m + 1$. The case $q = m$ can be rewritten as

$$\sum_j \frac{\gamma_j}{q} = \frac{1}{m} \sum_j \sum_{l=\max\{1, s-n+j\}}^{\min\{m, j\}} \frac{\binom{j-1}{l-1}\binom{n-j}{s-l}}{\binom{n}{s}} = 1.$$

Thus for the case $q = m + 1$, we have

$$\begin{aligned}
\sum_j \frac{\gamma_j}{q} &= \frac{1}{m+1} \sum_j \sum_{l=\max\{1, s-n+j\}}^{\min\{m+1, j\}} \frac{\binom{j-1}{l-1}\binom{n-j}{s-l}}{\binom{n}{s}} \\
&= \frac{1}{m+1} \sum_j \left( \sum_{l=\max\{1, s-n+j\}}^{\min\{m, j\}} \frac{\binom{j-1}{l-1}\binom{n-j}{s-l}}{\binom{n}{s}} + \frac{\binom{j-1}{m}\binom{n-j}{s-m-1}}{\binom{n}{s}} \right) \\
&= \frac{1}{m+1} \left( m + \sum_{j=m+1}^{n-s+m+1} \frac{\binom{j-1}{m}\binom{n-j}{s-m-1}}{\binom{n}{s}} \right)
\end{aligned}$$

where the last equality follows that $\binom{j-1}{m}\binom{n-j}{s-m-1} > 0$ for $m+1 \le j \le n - s + m + 1$ else 0. The remain proof is to show $\sum_{j=m+1}^{n-s+m+1} \frac{\binom{j-1}{m}\binom{n-j}{s-m-1}}{\binom{n}{s}} = 1$.

Consider selecting a subset of size $s$ from the set $\{1, 2, \ldots, n\}$. Arrange the elements of the subset in increasing order: $a_1 \ge a_2 \ge \cdots \ge a_s$ Then $a_{m+1}$ is the $(m + 1)$-th largest element. Let $j = m + 1$, i.e., the $(m + 1)$-th largest element is at position $j$.

To construct such a subset, the following conditions must be satisfied:

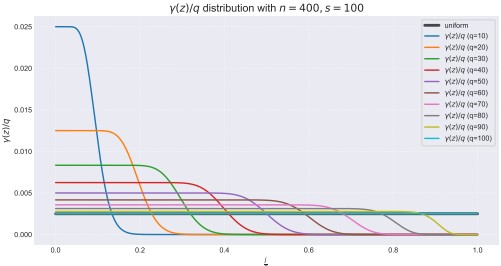 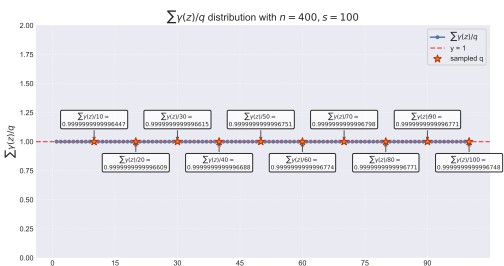

Figure 7: Normalized distributions $\gamma_j/q$ for different $q$ under $(n, s) = (400, 100)$. As $q$ increases, the curves flatten and approach the uniform distribution.

Figure 8: Empirical normalization of $\sum_{j=1}^{n} \gamma_j/q$. Across all $q \in \{10, \dots, 100\}$, the values remain extremely close to 1.

- Choose $m$ elements from the first $j - 1$ elements (i.e., $a_1, \dots, a_m$), which can be done in $\binom{j-1}{m}$ choices.

- Choose $s - m - 1$ elements from the remaining $n - j$ elements (i.e., $a_{m+2}, \dots, a_s$), which can be done in $\binom{n-j}{s-m-1}$ choices.

Thus, the number of subsets where the $m + 1$-th largest element is exactly at position $j$ is: $\binom{j-1}{m}\binom{n-j}{s-m-1}$

Summing over $j$ from $m + 1$ to $n - s + m + 1$ (since $j$ must be at least $m + 1$ and at most $n - s + m + 1$ to ensure enough elements remain), we obtain the total number of subsets of size $s$: $\sum_{j=m+1}^{n-s+m+1} \binom{j-1}{m}\binom{n-j}{s-m-1} = \binom{n}{s}$,

Therefore, the original claim holds for all $q$: $\sum_{j=1}^{n} \gamma_j/q = 1$. $\qquad\square$

### D.3 VALIDATION OF CONVERGENCE ASSUMPTIONS

To further support the validity of Theorem 3, we analyze whether the selected coreset stabilizes during training. Although the selection depends on sampling scores $H$, which may vary across epochs, our analysis shows that the coreset indeed becomes stable in later stages of training.

**Coreset Dynamics.** Our analysis does not assume a fixed coreset. Instead, OrderDP naturally determines the coreset through the pruning strategy (captured by $\gamma_j$ in Proposition 2), where the sample $z_j \in$ coreset follows an approximate Beta-distribution.

**Theoretical Parallel to SGD.** The applicability of Theorem 3 is analogous to SGD's convergence guarantees: (i) SGD converges by deterministic batches per epoch (the sample $z_j \in$ selected follows uniform sampling); (ii) OrderDP achieves convergence after the coreset stabilizes (via Beta-distributed sampling).

To empirically verify this stabilization, we measure the *Jaccard Similarity* between the coreset at the current epoch and that from the immediately preceding epoch, defined as

$$J(A, B) = \frac{|A \cap B|}{|A \cup B|}. \tag{40}$$

A higher similarity indicates that the selected set of samples remains consistent across epochs. Table 8 shows that OrderDP consistently achieves higher coreset stability than InfoBatch, confirming the practical applicability of Theorem 3.

### D.4 SAMPLE COVERAGE UNDER DIFFERENT PRUNING RATIOS

To further examine the exploration behavior of **OrderDP**, we track for each training sample the number of times it is selected into the update set throughout the entire training process. Figures 9, 10,

Table 8: Jaccard Similarity between consecutive checkpoints on CIFAR-100 with ResNet-18. InfoBatch and OrderDP are trained for 200 epochs (checkpoints at 20%, 40%, 60%, 80%, and final 100%), with learning rate = 0.03 and batch size = 128. Each setting is repeated 5 times, and mean ± std are reported. Higher values indicate greater stability of the coreset.

| Prune Ratio | Method | 0–20% | 20–40% | 40–60% | 60–80% | 80–100% | 100% |
|---|---|---|---|---|---|---|---|
| 40% | InfoBatch | **0.583±0.050** | 0.496±0.010 | 0.479±0.003 | 0.481±0.006 | 0.512±0.010 | 0.522±0.007 |
| | OrderDP | 0.645±0.057 | **0.692±0.023** | **0.713±0.008** | **0.743±0.023** | **0.757±0.034** | **0.767±0.008** |
| 70% | InfoBatch | **0.593±0.012** | 0.532±0.024 | 0.459±0.019 | 0.403±0.018 | 0.455±0.036 | 0.512±0.012 |
| | OrderDP | 0.552±0.049 | **0.592±0.016** | **0.647±0.020** | **0.661±0.018** | **0.678±0.023** | **0.704±0.090** |

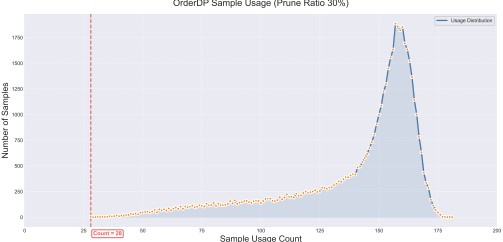

Figure 9: Sample usage count distribution (30% prune ratio).

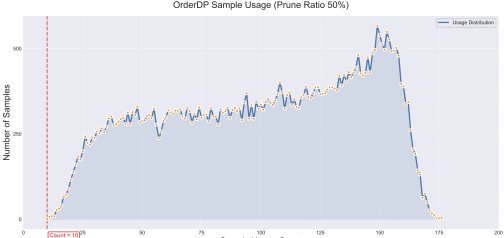

Figure 10: Sample usage count distribution (50% prune ratio).

and 11 show the empirical distributions of sample usage counts on CIFAR-10 under prune ratios of 30%, 50%, and 99%, respectively.

Across all pruning ratios, we observe that *no sample has zero usage count*: every example is selected at least once during training. Under practical pruning settings (e.g., 30%–50%), most samples fall within a reasonably concentrated range of usage counts, indicating that **OrderDP** does not permanently discard any data point but instead explores the entire dataset with a frequency controlled by $(s, q)$.

These empirical findings are fully consistent with our theoretical analysis of coverage and directly support our response to reviewer questions regarding whether OrderDP eventually sees the entire dataset.

### D.5 TIME-TO-ACCURACY CURVES

To complement the wall-clock results in Figure 3 and to directly address the reviewer's suggestion on evaluating time-to-accuracy, we report curves showing the relationship between training time and accuracy. These curves provide a practical view of how fast different methods reach comparable accuracy levels in real training scenarios.

Figures 12 and 13 present the Time-to-Accuracy curves on CIFAR-10 using ResNet-18 under prune ratios of 40% and 70%. Consistent with our findings throughout the paper, **OrderDP** achieves faster accuracy improvement and maintains stable convergence compared with both InfoBatch and Random, especially under higher pruning levels.

### D.6 STABILITY ANALYSIS

We further include a stability study of dynamic pruning methods under multiple independent runs. In this analysis, we evaluate CIFAR-100 with ResNet-18 at a 70% real pruning ratio across 10 runs. InfoBatch often exhibits large mid-training gradient oscillations and occasional convergence failures, while OrderDP consistently converges smoothly in every trial. Moreover, InfoBatch relies on late-stage full-data "annealing" to stabilize training, whereas OrderDP maintains exact pruning control via $(s, q)$ without requiring such rescue. This highlights the robustness and practicality of our approach. Under aggressive pruning, InfoBatch's rescaling further causes severe fluctuations in both gradient and accuracy.

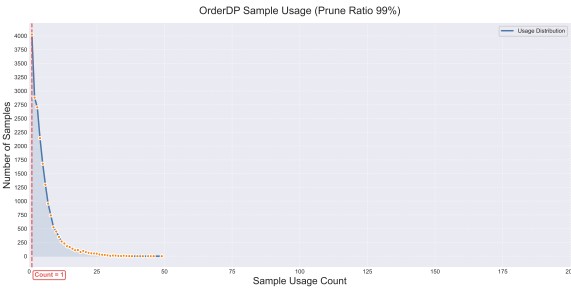

Figure 11: Sample usage count distribution under a 99% prune ratio. Even under extreme pruning, every sample is selected at least once.

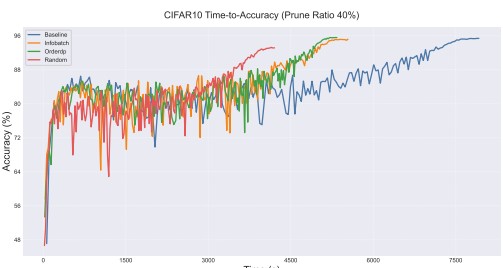

Figure 12: Time-to-Accuracy on CIFAR-10 at 40% prune ratio.

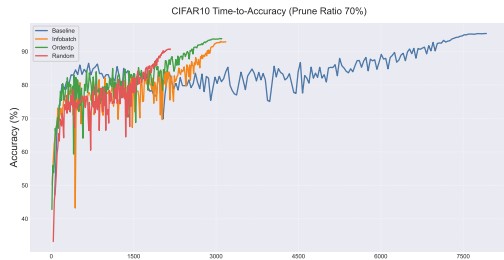

Figure 13: Time-to-Accuracy on CIFAR-10 at 70% prune ratio.

Detailed results are summarized in Table 9 (real prune ratio $\approx 0.61$, InfoBatch nominal prune ratio $\approx 0.99$; 200 epochs; learning rate = 0.03; batch size = 128; averaged over 10 runs).

# E  EXTENDED ANALYSIS OF GRADIENT BIAS

## E.1  MECHANISMS AND TOY EXAMPLE

This subsection provides additional clarification on how OrderDP eliminates biased gradient estimation. The method addresses the issue through three main mechanisms. **Uniform exploration.** Before pruning, $s$ points are randomly sampled so that every sample, regardless of gradient magnitude, has equal probability of entering the coreset. This ensures the gradient distribution is much closer to that of full-data SGD compared with InfoBatch's biased rescaling strategy. **Unbiased surrogate loss.** Instead of pruning the original loss directly, a new loss $\mathcal{L}_q$ is constructed with closed-form weights $\gamma_j$ derived from a two-stage sampler. Theorem 1 guarantees each update is an unbiased estimator of $\nabla\mathcal{L}_q(\theta)$, while Theorems 2, 3, and 4 provide convergence and generalization guarantees, avoiding the need for late-stage full-data annealing. **Exact pruning control.** By explicitly setting $(s, q)$, any target prune ratio (e.g., 70%, 80%, 90%) can be realized precisely, unlike InfoBatch's mean-threshold scheme, which fluctuates around $\sim 77\%$.

To make the difference clear, we present a toy comparison under an 80% prune ratio with 5 samples of gradients $g = \{1, 2, 3, 4, 5\}$. In **InfoBatch (mean-threshold + rescale)**, samples 3, 4, 5 are always kept, while 1 and 2 are included with probability 0.2, and their gradients rescaled by a factor $1/(1 - 0.8) = 5$. This leads to four possible outcomes summarized in Table 10.

From Table 10, the expected gradient estimate is $4.336$ with prune ratio $< 0.8$, indicating bias.

In contrast, for **OrderDP (5 samples, 80% prune)**, we randomly sample a batch with size $s \in \{1, \ldots, 5\}$ and select the top-1 element. The detailed probability calculation for each index is as follows:

Table 9: Stability comparison on CIFAR-100 with ResNet-50 across 10 runs under varying **training progress** (percentage of epochs). Reported are **Accuracy (% ± Std)** and **Gradient Std**. OrderDP shows smoother convergence and eliminates instability observed in InfoBatch.

| Method | Metric | 0–30% | 30–50% | 50–70% | 70–100% | Final |
|---|---|---|---|---|---|---|
| $\epsilon$-Greedy | Acc ± Std | 48.01 ± 4.60 | 49.77 ± 2.54 | 52.61 ± 1.42 | 66.24 ± 1.23 | 74.77 ± 0.30 |
| | Grad ± Std | 3.08 ± 1.19 | 3.49 ± 0.62 | 2.78 ± 0.57 | 2.05 ± 0.45 | 1.53 ± 0.37 |
| UCB | Acc ± Std | 49.70 ± 4.80 | 50.66 ± 2.31 | 54.34 ± 1.82 | 67.97 ± 1.12 | 75.41 ± 0.40 |
| | Grad ± Std | 4.08 ± 1.69 | 3.69 ± 1.02 | 2.99 ± 0.27 | 2.35 ± 0.38 | 1.33 ± 0.14 |
| InfoBatch | Acc ± Std | 45.74 ± 3.56 | 52.08 ± 2.55 | 47.72 ± 3.63 | 68.92 ± 3.01 | 76.72 ± 0.70 |
| | Grad ± Std | 7.35 ± 1.78 | 5.88 ± 1.24 | 4.35 ± 9.48 | 3.56 ± 4.55 | 2.89 ± 1.67 |
| OrderDP | Acc ± Std | 48.00 ± 3.23 | 56.00 ± 2.01 | 61.00 ± 1.34 | 72.00 ± 0.56 | 78.32 ± 0.20 |
| | Grad ± Std | 4.08 ± 1.19 | 3.49 ± 0.62 | 2.78 ± 0.47 | 2.05 ± 0.55 | 1.03 ± 0.33 |
| Whole Dataset | Acc ± Std | 56.58 ± 1.56 | 62.36 ± 0.76 | 66.84 ± 0.52 | 72.24 ± 0.40 | 80.60 ± 0.20 |
| | Grad ± Std | 3.88 ± 0.79 | 3.09 ± 0.41 | 2.45 ± 0.42 | 1.93 ± 0.55 | 0.88 ± 0.20 |

Table 10: Toy example of InfoBatch under an 80% prune ratio with 5 gradients. The table shows the kept set, probability of selection, rescaled gradients, average gradient, and prune rate.

| Kept set | Probability | Gradients | Avg. grad. | Prune rate |
|---|---|---|---|---|
| $\{3, 4, 5\}$ | 0.64 | $3, 4, 5$ | 4.0 | 0.60 |
| $\{1, 3, 4, 5\}$ | 0.16 | $5 \cdot 1, 3, 4, 5 = 5, 3, 4, 5$ | 4.25 | 0.20 |
| $\{2, 3, 4, 5\}$ | 0.16 | $5 \cdot 2, 3, 4, 5 = 10, 3, 4, 5$ | 5.5 | 0.20 |
| $\{1, 2, 3, 4, 5\}$ | 0.04 | $5 \cdot 1, 5 \cdot 2, 3, 4, 5 = 5, 10, 3, 4, 5$ | 5.4 | 0.00 |

$$\{1\}: \quad \tfrac{1}{5} \cdot \tfrac{1}{5} \, (s = 1) = \tfrac{1}{25},$$

$$\{2\}: \quad \tfrac{1}{5} \cdot \tfrac{1}{5} \, (s = 1) + \tfrac{1}{10} \cdot \tfrac{1}{5} \, (s = 2) = \tfrac{3}{50},$$

$$\{3\}: \quad \tfrac{1}{5} \cdot \tfrac{1}{5} \, (s = 1) + \tfrac{3}{10} \cdot \tfrac{1}{5} \, (s = 2) + \tfrac{1}{10} \cdot \tfrac{1}{5} \, (s = 3) = \tfrac{3}{25},$$

$$\{4\}: \quad \tfrac{1}{5} \cdot \tfrac{1}{5} \, (s = 1) + \tfrac{3}{10} \cdot \tfrac{1}{5} \, (s = 2) + \tfrac{3}{10} \cdot \tfrac{1}{5} \, (s = 3) + \tfrac{1}{10} \cdot \tfrac{1}{5} \, (s = 4) = \tfrac{9}{50},$$

$$\{5\}: \quad \tfrac{1}{5} \cdot \tfrac{1}{5} \, (s = 1) + \tfrac{2}{5} \cdot \tfrac{1}{5} \, (s = 2) + \tfrac{3}{5} \cdot \tfrac{1}{5} \, (s = 3) + \tfrac{4}{5} \cdot \tfrac{1}{5} \, (s = 4) + \tfrac{1}{5} \, (s = 5) = \tfrac{3}{5}.$$

The expected gradient estimate under this distribution is $4.06$ with prune ratio exactly $0.8$. Therefore, OrderDP not only maintains the target pruning ratio precisely but also achieves gradient estimates closer to the true full gradient ($= 3$), effectively eliminating the bias observed in InfoBatch.

### E.2 GRADIENT DIRECTION ANALYSIS

In addition to gradient magnitude analysis, we also examine gradient directions by measuring the *cosine similarity* between the gradients computed with each pruning method and the full-data gradient at matched checkpoints (same model weights). We train on CIFAR-100 with ResNet-18 for 200 epochs, evaluate at 20%, 40%, 60%, 80%, and 100% of training progress, using a learning rate of 0.03 and batch size of 128. Each setting is repeated 5 times, and we report mean ± std. Results under pruning ratios 40% and 70% are summarized in Table 11.

These results demonstrate that **OrderDP's gradients align more closely with full-data gradients**, particularly at high pruning ratios, thereby reducing directional bias compared with InfoBatch.

## F LIMITATIONS AND FUTURE WORK

While `OrderDP` excels on moderate-scale vision benchmarks, its performance on very large architectures, streaming inference scenarios, and heterogeneous hardware platforms, as well as in self-supervised or multi-modal settings, remains to be explored. In future work, we will extend `OrderDP` to adaptive pruning schedules, investigate its integration with transformer and graph models, and study its behavior under distribution shift and noisy labels.

Table 11: Cosine similarity between pruned and full-data gradients on CIFAR-100 (ResNet-18) under pruning ratios 40% and 70%. Each experiment is repeated 5 times, and mean $\pm$ std are reported. Higher values indicate stronger alignment with full-data gradients.

| Prune Ratio | Method | 0–20% | 20–40% | 40–60% | 60–80% | 80–100% | 100% |
|---|---|---|---|---|---|---|---|
| 40% | InfoBatch | 0.915±0.044 | 0.940±0.008 | 0.916±0.007 | 0.904±0.011 | 0.897±0.008 | 0.895±0.009 |
| | OrderDP | **0.943±0.035** | **0.951±0.007** | **0.934±0.008** | **0.921±0.009** | **0.908±0.014** | **0.906±0.011** |
| 70% | InfoBatch | 0.825±0.037 | 0.807±0.027 | 0.763±0.021 | 0.758±0.023 | 0.743±0.026 | 0.716±0.029 |
| | OrderDP | **0.853±0.048** | **0.896±0.018** | **0.901±0.015** | **0.893±0.014** | **0.868±0.021** | **0.844±0.018** |

Another promising direction is to develop noise-robust variants of `OrderDP`. Although the current work focuses on the top-$q$ strategy, the surrogate-loss framework introduced in this paper is more general and can naturally incorporate min-$q$ selection or mixed hard/easy sampling schemes by modifying the weight structure $\{\gamma_j\}$. Such extensions may help suppress extreme outliers, improve stability under label noise, and adapt the pruning strategy across different stages of training (e.g., hard-sample emphasis in early stages and easy-sample regularization in later stages). We plan to further explore these variants and evaluate their performance in noisy or adversarial settings.

## G    THE USE OF LARGE LANGUAGE MODELS (LLMS)

Large language models (LLMs) were used solely for linguistic refinement and editing of the manuscript. All scientific ideas, methodological contributions, and experimental results are entirely conceived, implemented, and validated by the authors.

