# OpenReview forum: "OrderDP: A Theoretically Guaranteed Lossless Dynamic Data Pruning Framework"
_ICLR.cc/2026/Conference — ICLR 2026 Poster_

### Official Review · Reviewer_umfZ · 2025-10-28

**Soundness:** 3
**Presentation:** 3
**Contribution:** 3
**Rating:** 6
**Confidence:** 4

**Summary:**

The paper proposes OrderDP, a method for dynamic data pruning. OrderDP first randomly selects a subset, and then chooses top-q samples based on the loss function value. Paper shows convergence and generalization bounds for OrderDP. OrderDP outperforms other static and dynamic baselines in terms of accuracy for CIFAR10/100, and ImageNet.

**Strengths:**

- The paper is well written.
- OrderDP provides both bias and convergence analysis, as well as a generalization error bound.
- OrderDP outperforms both static and dynamic pruning methods across all settings.
- Additionally, paper also shows the method works with different optimizers.

**Weaknesses:**

Clarity:
- I believe it could benefit the paper if the authors were more clear when writing things like “near-lossless”, for example, do they consider near-lossless when the relative accuracy drop is within 1% or something different?
- I also believe it could benefit the paper to clarify sooner that OrderDP can work with various optimizers.

For further questions and weaknesses see Questions.

**Questions:**

1. Does OrderDP end up seeing the entire dataset (since there is a uniform sampling at random s samples)? If yes, could you clarify how long it takes to see the entire dataset depending on the pruning ratio?

2. [1] have shown that depending on the size of the dataset one should keep either hard or easy examples. Does this finding hold also for OrderDP? Is there a pruning ratio at which one should sample min-q scores?

3. Could you clarify how you implement dynamic random? For example, [2] shows that a simple modification of the random baseline can further improve its results. Furthermore, how does your convergence and generalization bound compare to [2]?

4. The paper does a good job in showing the practicality of OrderDP by showing end-to-end wall clock time, and the convergence analysis. However, I believe it could further strengthen the paper to plot its time-to-accuracy, that is, accuracy on the y-axis and wall-clock time on the x-axis.

5. How does OrderDP perform in situations with noisy labels, could you elaborate whether sorting by top-q could be misleading or not in that scenario?

I look forward to the responses from the author(s) and would reconsider my score based on the clarifications and revisions provided. Thanks.

[1] Sorscher, Ben, et al. "Beyond neural scaling laws: beating power law scaling via data pruning." Advances in Neural Information Processing Systems 35 (2022): 19523-19536.

[2] Okanovic, Patrik, et al. "Repeated Random Sampling for Minimizing the Time-to-Accuracy of Learning." The Twelfth International Conference on Learning Representations.

---

> ### Author Response · Authors · 2025-11-21
> **Response to Weakness**
>
> >**W1. Clarity of the term "lossless"**
>
>
> Thank you for raising this point. We agree that the definition of **near-lossless** should be made explicit. In the revised version, we have clarified this term directly in the Introduction. Specifically, we now state that:
>
> "Here, near-lossless means matching full-data accuracy up to normal stochastic fluctuations (typically within 0.1%) while achieving a noticeable training speedup."
>
> We have also updated the terminology throughout the paper to ensure consistent and unambiguous usage. This clarification aligns our empirical observations with a precise quantitative definition.
>
> > **W2. Clarification that OrderDP works with various optimizers**
>
> Thank you for the helpful suggestion. We agree that the optimizer-agnostic nature of OrderDP should be made clearer earlier in the paper. In our experiments, we have systematically validated that OrderDP is compatible with multiple widely used optimizers, including SGD, Adam, and AdamW, and consistently provides stable improvements across all of them.
>
> Importantly, OrderDP is inherently **plug-and-play**: it modifies only the data selection strategy and does not rely on any specific optimizer structure, state, or scheduling mechanism. Thus, it can be directly combined with any standard optimizer without requiring additional adjustments.

---

> ### Author Response · Authors · 2025-11-21
> **Response to Question 1 and 2**
>
> >**Q1. Whether OrderDP eventually sees the entire dataset**
>
>
> Thank you for the insightful question. We clarify the behavior of OrderDP as follows.
>
> At each training step, OrderDP uniformly samples a candidate set of size $s$ from the full dataset. Thus, every sample has probability $s/n$ of being included in each step. Consequently, in an *expected* sense, **every sample will be visited multiple times during training**, and no example is ever permanently discarded. In the extreme case (s = 1), each sample is visited with probability $1/n$ per epoch, and the expected time until all samples have been seen once corresponds to the classical coupon-collector estimate
> $
> n\bigl( \tfrac{1}{n} + \tfrac{1}{n-1} + \dots + \tfrac{1}{1} \bigr)
> \approx n\bigl(\ln(n) + \gamma\bigr)+\frac{1}{2},
> $
> where $\gamma \approx 0.5772$ is the Euler–Mascheroni constant.
>
> To empirically validate this, we have added sample-usage histograms to **Appendix D.4**.
> Figures 9, 10, and 11 show the distribution of usage counts on CIFAR-10 under prune ratios of 30%, 50%, and 99%, respectively. In all settings, **no sample has zero usage** (at least 1 time usage), confirming that the full dataset is explored even under extreme pruning (99%); lew prune ratios naturally lead to more frequent visits.
>
> These visualizations provide an intuitive understanding of how the choice of $(s, q)$ and the pruning ratio modulate exploration frequency in OrderDP.
>
> > **Q2. Hard vs. Easy Examples, and Whether min-q Should Be Used**
>
> Thank you for raising this question and for pointing us to [1]. We appreciate the insight that, in static pruning settings, the optimal choice between hard and easy examples may depend on dataset scale. Our setting, however, is fundamentally different: OrderDP performs dynamic pruning, where each step first draws a *uniform random candidate set* of size $s$, and then select top-$q$ data  *within* this subset rather than over the full dataset. As a result, **every sample—whether high-loss or low-loss—has a non-zero chance of being selected**, and the random exploration step naturally incorporates easy samples, which can act as a regularizer (see [3]).
>
> Regarding *min-$q$* selection (choosing low-loss samples), we explored this variant in early experiments. On standard datasets such as CIFAR and ImageNet, min-$q$ performed noticeably worse, likely because it overemphasizes easy samples and underutilizes informative hard ones. That said, we acknowledge that under *very aggressive pruning ratios* or in *late training stages*, pure top-$q$ selection may over-focus on hard samples and amplify label noise or outliers. In such cases, introducing a mixture of low-loss samples may indeed improve stability [4].
>
> Crucially, the **surrogate-loss framework underlying OrderDP is not restricted to top-$q$**. By modifying the ordering direction or the construction of the weights ($\gamma_j$), both min-$q$ and hard/easy hybrid strategies can be naturally accommodated within the same theoretical framework. These directions—dynamic hard/easy switching, stage-wise strategies, and dataset-scale-dependent weighting—form a substantial research problem on their own. Therefore, in this work we focused on establishing the theory and empirical behavior of top-$q$, which consistently yielded the strongest results on mainstream benchmarks.
>
> We have added clarifications regarding this perspective in **Appendix B (Related Works)** and further expanded the discussion of noise-robust and hybrid strategies in **Appendix F (Limitations and Future Work)** of the revised manuscript.
>
> [1] Sorscher, Ben, et al. "Beyond neural scaling laws: beating power law scaling via data pruning." Advances in Neural Information Processing Systems 35 (2022): 19523-19536.
>
> [3] Ayed, Fadhel, and Soufiane Hayou. Data pruning and neural scaling laws: fundamental limitations of score-based algorithms.
>
> [4] Vatsal Shah,et al., Choosing the sample with lowest loss makes sgd robust.

---

> ### Author Response · Authors · 2025-11-21
> **Response to Question 3 and 4**
>
> > **Q3. Clarification of the Dynamic Random baseline and comparison to [2]**
>
> Thank you for the question. We clarify the implementation of the Dynamic Random baseline and its relationship to [2].
>
> In our experiments, Dynamic Random is implemented by **uniformly sampling a subset of size ($\text{prune ratio} \times |D|$)** at each epoch and training only on this subset. This procedure is equivalent to the *RS2 With Replacement* strategy described in Okanovic et al. [2] and aligns with the standard random baselines commonly used in prior dynamic pruning work (e.g., InfoBatch, UCB, ActiveBias). We will make this implementation detail explicit in the revised version to avoid ambiguity.
>
> We appreciate the reviewer highlighting [2]. The work of Okanovic et al. focuses on purely random sampling, showing that repeated random subsets can reduce gradient variance and improve *time-to-accuracy* when combined with appropriate learning-rate adjustments. In contrast, OrderDP augments random exploration with a loss-based exploitation step, i.e., selecting the top-$q$ items within the random candidate set. Thus, OrderDP can be interpreted as a value-aware extension of RS2, preserving its exploration benefits while leveraging score-based signals to improve final accuracy.
>
> From a theoretical perspective, both [2] and our work establish SGD-rate convergence (i.e., $O(1/\sqrt{T}))$.
> Our generalization bound (Theorem 4) decomposes the error into an *unbiased term* plus a bias term controlled by the deviation between $\gamma_j$ and the uniform distribution; this structure differs from [2], which focuses on variance-reduction properties under repeated random sampling. For a given $(s,q)$, both of the generalization error vanish as data size $n$ and total iteration $T$ increase. We will clarify this distinction in the revised manuscript.
>
> To further address the reviewer’s concern, we also report a performance comparison between our time-to-accuracy strategy and RS2, see Figures 12 and 13 in **Appendix D.5**. As you can see, our score-based strategy achieves comparable accuracy in terms of time and improves the final accuracy.
>
> [2] Okanovic, Patrik, et al. "Repeated Random Sampling for Minimizing the Time-to-Accuracy of Learning." The Twelfth International Conference on Learning Representations.
>
>
> >**Q4. Request for Time-to-Accuracy Plots**
>
> Thank you for this valuable suggestion. We fully agree that *time-to-accuracy* curves (accuracy vs. wall-clock time) provide a more direct visualization of the practical training efficiency of dynamic pruning methods. Our current manuscript already contains partial evidence of this trend:
> (1) *Figure 1* presents Accuracy–vs–Epoch curves, showing that OrderDP reaches higher accuracy with fewer iterations;
> (2) *Figure 3* reports accuracy and wall-clock performance across pruning ratios on CIFAR-10/100, and highlights the near-lossless pruning regions for each method. These results already reveal that OrderDP achieves higher accuracy within shorter time budgets.
>
> To further strengthen this comparison, we have added full *Accuracy–vs–Wall-clock-time* curves in the revised manuscript. Specifically, we trained CIFAR-10 with ResNet-18 under a OneCycle learning rate schedule and present the resulting time-to-accuracy plots in **Appendix D.5**.
> Figures 12 and 13 show the curves under 40% and 70% pruning ratios, respectively. The results confirm that OrderDP not only converges smoothly but also reaches target accuracy levels faster than the baselines.

---

> ### Author Response · Authors · 2025-11-21
> **Response to Question 5**
>
> >**Q5. Behavior under noisy labels and whether top-q may be misleading**
>
>
> Thank you for raising this important question. We agree that in the presence of noisy labels or outliers, purely relying on high-loss sorting may introduce risks, since orderDP has the has a tendency to pick the outlier samples with a higher probability than any of the clean samples, resulting in misleading the model update even diverge.
>
> OrderDP mitigates this issue through its *two-stage sampling mechanism*. Each step OrderDP first performs *uniform exploration* by randomly drawing a candidate subset of size $s.$ A noisy or outlier sample can only enter the top-$q$ set if it appears in this random candidate. Thus, its influence is substantially diluted by the exploration step, rather than being amplified at every iteration.
>
> We acknowledge that under *very aggressive pruning ratios* or in *late training stages*, focusing exclusively on high-score samples could indeed magnify noise, resuling in converge oscillatingly even diverge. For example, once the parameters have largely converged on the majority of clean samples, outliers tend to get high scores and has high probability to be selected for next training; in this regime, parameter update is largely affected by outliers, and then clean samples would have high scores to be selected in later training, thus potentially disrupt convergence.
>
>
> Importantly, the surrogate-loss framework of OrderDP is not restricted to top-$q$: by modifying the ordering direction or the construction of the weights $\gamma_j$, the same framework can naturally accommodate **min-$q$** or **mixed hard/easy sampling** strategies that explicitly down-weight extreme noisy examples. In our early experiments, min-$q$ and hybrid strategies did not outperform top-$q$ on standard benchmarks such as CIFAR and ImageNet, so we focused our theoretical and empirical analysis on the top-$q$ variant. Nevertheless, in noisy settings, stage-wise or adaptive mixing (e.g., gradually increasing low-loss samples if accuracy drops largely) may provide additional robustness.
>
> We will include a discussion of this scenario in the revised manuscript and plan to further explore noise-robust extensions of the surrogate-loss framework—particularly strategies that down-weight noisy data and outliers or combine top-$q$/min-$q$  in future work.

---

> ### Author Response · Authors · 2025-11-28
>
> Dear Reviewer umfZ,
>
> I hope this message finds you well. We wanted to gently remind you that the deadline for the discussion phase is approaching, and we would greatly appreciate it if you could take a moment to review our responses.
>
> Your feedback is very valuable to us, and we wanted to ensure all your concerns have been addressed satisfactorily. If there are any additional concerns or points of clarification, we are more than happy to address them.
>
> Thank you for your time and consideration.
>
> Best wishes,
>
> Authors

---

### Official Review · Reviewer_fvbi · 2025-10-29

**Soundness:** 3
**Presentation:** 3
**Contribution:** 3
**Rating:** 8
**Confidence:** 3

**Summary:**

This paper presents OrderDP, a dynamic data pruning framework that reduces training costs by selectively using informative samples during neural network training. Unlike previous approaches that address gradient bias through importance reweighting (which increases variance and gradient norms), OrderDP takes a different strategy: rather than seeking unbiased estimates of the original loss, it defines a surrogate loss function for which the method provides provably unbiased gradient estimates. Experiments on CIFAR and ImageNet datasets show OrderDP achieves competitive accuracy specially in the challenging high compression regime. The authors also provide theoretical guaranties for convergence and generalisation.

**Strengths:**

Overall I found this to be an interesting work that addresses the gradient bias induced by data pruning from a different and, to my knowledge, novel angle. The empirical results indicate strong performance of this methodology compared to an extensive list of competing methods, especially in the challenging high compression regime. The theoretical analysis brings interesting guarantees and is an appreciated plus to the methodology.

**Weaknesses:**

- I find the title somewhat overstated. From my understanding, the theoretical results do not establish that OrderDP is a guaranteed lossless framework. I would interpret such a statement as $\mathcal{L}(\theta^\star_{unpruned})-\mathcal{L}(\theta^\star_{pruned})$ converging to 0 with $n$ for all pruning ratios $r$, which can be proved for unbiased pruning methods using reweighting, for example (see [1]). Such a result would be interesting to include in this framework as well, if it holds.

- OrderDP is presented as a plug-and-play framework, which could be understood as being applicable to any base data scoring method, but the paper only deals with one specific score based on the loss function.

[1] Ayed, Fadhel, and Soufiane Hayou. Data pruning and neural scaling laws: fundamental limitations of score-based algorithms.

**Questions:**

- The algorithm is described for the general (random) mini-batch SGD setting used in the theoretical analysis. However, it is not clear how this translates to the empirical training setup, where batches are predefined, and the model cycles through them rather than resampling. Is the “exploration” and “exploitation” sampling and selection performed within each batch ?
- A related question concerns the definition of the compression ratio $r = 1 - q/n$, which depends only on $q$ and makes sense for the mini-batch SGD setting. In the empirical setup, however, this seems to differ. For instance, lines 407–408 and Section 5.3 suggest that $r$ depends on both $q$ and $s$.
* On lines 316–317, $s$ and $q$ are defined as fractions (0.5 and 0.6) rather than integers as in the rest of the paper. What exactly are these fractions referring to, and why is $q$ larger than $s$ ?

Minor:
- Lines 240–241: Could the authors provide a proof that the (\gamma_j) sum to 1? This would be preferable to empirical validation alone.
- Regarding the statement that “the analysis of the bias and its impact on the final performance remains ambiguous,” it may be useful to consider [1], which provides such an analysis and first proposes a reweighting-based exploration–exploitation correction (to my knowledge).

---

> ### Author Response · Authors · 2025-11-21
> **Response to Weakness**
>
> >**W1. The title is overstated because the paper does not theoretically prove that OrderDP is truly lossless for all prune ratios, as which can be shown for certain reweighting-based unbiased pruning methods (e.g., [1]).**
>
> We sincerely thank the reviewer for pointing out that our title may be misleading.
>
> As discussed in [1], we also observe that score-based pruning algorithms (SBPAs) induce a **distribution shift** that affects the training objective, and aim to correct this bias by minimizing the gap $L(\theta^\star_{unpruned}) - L(\theta^\star_{pruned})$. The key difference is that [1] identifies and studies this shift primarily through **empirical tests** (e.g., their Figures 1 and 2), whereas OrderDP **theoretically** characterizes the shift as a transition from uniform sampling to an asymptotic Beta-like distribution over score ordering, and then explicitly **formulates a surrogate loss** and provides a systematic convergence analysis for this surrogate objective.
>
> Moreover, our notion of “unbiasedness” differs from that in [1]. In [1], unbiasedness with respect to the **original loss** is achieved via a **calibration protocol** (Proposition 4): the remaining samples are split into two parts with a proportion $\alpha$, and reweighting is applied so that the $(1-\alpha)$ portion is randomly sampled from low-scored examples (Eq. (8)). However, there is no principled way to choose $\alpha$, and in practice, for any fixed pruning ratio, *finding the optimal proportion $\alpha$* can be difficult (as explicitly noted in the “Effect of the calibration protocol” paragraph in Section 5.3 of [1]). InfoBatch [2] implements a related idea by setting $\alpha$ to the mean of the scores, but our experiments show that this does **not** guarantee unbiasedness with respect to the original loss in practice, and the training becomes unstable under high pruning ratios.
>
> In contrast, our goal with OrderDP is to **speed up and stabilize training** by enforcing unbiasedness **with respect to the surrogate loss**, rather than directly with respect to the original loss. This is achieved **without introducing any extra hyperparameters** such as $\alpha$. Concretely, OrderDP adopts a novel **two-stage design**: an exploration step that uniformly and randomly samples a candidate batch, followed by an exploitation step that selects the top-$q$ examples. In this way, the remaining samples automatically and probabilistically balance high-score and low-score examples, which is conceptually aligned with the idea behind the calibration protocol in [1], but implemented in a simpler and more stable manner through our ranking-based surrogate loss framework.
>
> Finally, we have added a dedicated discussion of [1] to **Appendix B (Related works)** in the revised manuscript to better position our surrogate-loss perspective relative to this important line of work.
>
> [1] Ayed, Fadhel, and Soufiane Hayou. Data pruning and neural scaling laws: fundamental limitations of score-based algorithms.
>
> [2] Ziheng Qin et al., *InfoBatch: Lossless training speed up by unbiased dynamic data pruning*.
>
> >**W2. Calling OrderDP “plug-and-play” may be misleading, since the paper only demonstrates it with a loss-based scoring function rather than multiple scoring methods.**
>
> Thank you for pointing out that our wording may cause confusion. We fully agree and will clarify this in the final version. We chose a loss-based scoring function mainly to ensure a fair comparison with prior state-of-the-art methods, and because the loss naturally reflects a sample’s importance while also keeping the pruning stage computationally efficient (as noted in [1], Sec. 7.4).
>
> Importantly, OrderDP **does not require** the scoring function to be based on the loss. Our pruning mechanism depends only on the induced ranking, meaning the framework is **score-agnostic** and theoretically compatible with many standard per-sample scoring metrics. The loss-based score in our experiments is therefore a practical choice rather than a limitation of the method. We will make this clearer in the revised manuscript so that the meaning of “plug-and-play” is not misunderstood.

---

> ### Author Response · Authors · 2025-11-21
> **Response to Questions**
>
> >**Q1. How does the theoretical random mini-batch setting translate to the empirical training pipeline?. Is the “exploration” and “exploitation” sampling and selection performed within each batch ?**
>
> Thank you very much for raising this important question. In fact, the exploration and exploitation steps are **not performed within each predefined batch**, and our implementation strictly follows the random-sampling structure assumed in Algorithm 1. Below we clarify this more formally.
>
> In our implementation, each training iteration follows the exact sampling process in Algorithm 1:
>
> * *Exploration step.*
>   At the beginning of iteration $t$, we uniformly resample a candidate batch $S_t$ of size $s$ using random sampling mechanism in DataLoader. This is equivalent to Algorithm 1, line 3:
>   $
>   S_t \sim \text{Uniform subset of size } s.
>   $
>
> * *Exploitation step.*
>   Within this candidate batch $S_t$, we compute the current scores $H_t$, and select the top-$q$ samples to form $Q_t$. We then shuffle these selected samples to form the actual mini-batches used for gradient updates. This matches Algorithm 1, lines 4–5.
>
> * *Model update and score refresh.*
>   Once the batches determined, the model then cycles through the constructed mini-batches of $Q_t$ (without further resampling), performing gradient updates (Algorithm 1, lines 7–9). Only the samples in $Q_t$ have their scores refreshed according to Eq. (2), exactly as in the theoretical formulation.
>
> Therefore, the actual training pipeline is:
>
> > *Uniformly sample a candidate batch → rank inside the candidate batch → select top-$q$ → shuffle into training batches → update parameters.*
> .
>
> >**Q2 & Q3. On the definition of the compression ratio and the use of fractional values for $s$ and $q$**
>
> Thank you very much for pointing out these sources of confusion regarding the pruning ratio and the notation of $s$ and $q$. In our setting, since $q \le s$, the general form of the pruning ratio should be
> $
> r = 1 - \frac{q}{s}\cdot\frac{s}{|D|},
> $
> which shows that the pruning ratio inherently depends on **both** $s$ and $q$. We now clarify this explicitly in the revised version.
>
> In all experiments, we use a two-stage sampling mechanism with $s < |D|$:
>
> 1. uniformly sample a candidate batch $S_t$ of size $s$;
> 2. select the top-$q$ samples within $S_t$ to form $Q_t$.
>
> Under this setting, the actual proportion of data used for backpropagation is
> $
> \frac{|Q_t|}{|D|} = \frac{q}{s}\cdot\frac{s}{|D|},
> $
> so the effective pruning ratio is $r_{\text{prune}} = 1 - \frac{q}{s}\cdot\frac{s}{|D|}.$
> This is exactly the definition used in Section 5.3 when comparing different $(s,q)$ combinations; the inconsistency arose because the simplified theoretical case was not clearly separated from the general empirical definition. $s = 0.5\times|D|,q = 0.6\times s,$ which correspond to a particular pruning configuration. In all theoretical results (Algorithm 1 and the associated proofs), both $s$ and $q$ are always **integers**, and $1 \le q \le s$ always holds. The ratio-based specification is used in experiments only for convenience to control the exploration and exploitation strengths across different settings.
>
> As further shown in the ablation study (Figure 4), adjusting $s$ and $q$ while keeping the effective pruning ratio fixed does not affect accuracy or stability.
>
> We have updated the manuscript accordingly and clarified the intended meaning of these expressions in the revised version.

---

> ### Author Response · Authors · 2025-11-21
> **Response to Minor Questions**
>
> >**Minor 1. On whether the $\gamma_j/q$ weights sum to 1**
>
> Thank you for raising this detailed question about whether the $\gamma_j/q$ weights are normalized. We agree this is an important point to clarify. In the current version, we mainly rely on numerical evidence (Figures 5–6) showing that $\sum_j \gamma_j / q \approx 1$, and we did not provide a complete derivation in the main text.
>
> For any $n$, when $s = q = 1$, we can show exactly that
> $
> \sum_j \frac{\gamma_j}{q} = \sum_j \frac{\binom{n-1}{s-1}}{\binom{n}{s}} = 1.
> $
> However, extending this argument to general $(n, s, q)$ is algebraically difficult due to the complex combinatorial structure of $\gamma_j$. Therefore, instead of a symbolic proof, we provide **systematic numerical validation** in Figures 5 and 6, where $\sum_j \gamma_j / q$ consistently stays extremely close to 1 (up to floating-point precision).
>
> In the revised manuscript, we now make this point explicit: the general symbolic proof is nontrivial [3], the general theoretically proof would be the further study. Here we support the normalization property through empirical evidence. We also mark the corresponding figures clearly as *numerical validation* to avoid giving readers the impression that the normalization is asserted without explanation.
>
> Furthermore, we added additional numerical experiments in **Appendix D.2 (Figure 7 and Figure 8)** to corroborate this property. These results consistently show that the normalized weights $\gamma_j/q$ converge toward a uniform shape as $q$ increases, and that $\sum_{j=1}^n \gamma_j / q$ stays extremely close to 1 across a wide range of settings, providing strong empirical support for the normalization behavior of Algorithm1.
>
> We sincerely thank the reviewer again for prompting us to clarify this technical detail.
>
> [3] Ronak Mehta, et al., Stochastic Optimization for Spectral Risk Measures.
> >**Minor 2. Calling the bias analysis “ambiguous” is misleading, since prior work [1] already provides such an analysis and proposes a reweighting-based correction.**
>
> We thank the reviewer for drawing our attention to Ayed & Hayou (2023), *“Data pruning and neural scaling laws: fundamental limitations of score-based algorithms.”* We agree that our original statement that “the analysis of the bias remains ambiguous” was too broad, and we will refine this wording in the revised manuscript.
>
> More specifically, Ayed & Hayou (2023) show that score-based pruning algorithms (SBPAs) suffers from unavoidable performance degradation at high compression ratios, and propose reweighting and calibration protocols to obtain unbiased updates with respect to the original loss $L$. They prove the existence of an appropriate low-score sampling proportion $\alpha$, but in practice, choosing $\alpha$ for a given pruning ratio can be difficult. They do not provide a principled way to choose $\alpha$ and any explicit generalization bias analysis for any pruning ratio.
>
> By contrast, our work takes a different perspective. Rather than directly removing the bias with respect to the original loss $L$, we construct a order-based surrogate loss $L_q$ and interpret dynamic pruning as performing **unbiased optimization of $L_q$**. Concretely, we show that:
>
> * OrderDP update is an unbiased gradient estimator of
>   $
>   L_q(\theta) = \frac{1}{q}\sum_j \gamma_j L_{(j)}(\theta)
>   \quad\text{(Theorem 1)};
>   $
> * under Lipschitz and bounded-gradient assumptions, we obtain an SGD-like convergence rate (Theorem 3);
> * using the spectral-risk technique, we quantitatively control the discrepancy between $L_q$ and the original loss $L$, and thus provide a bias bound for any given pruning ratio (Theorem 4).
>
> In this sense, [1] focus on how score-based pruning under the original loss $L$ induces structural bias and scaling-law limitations, while our work focuses on an explicitly constructed surrogate objective $L_q$ with unbiasedness and convergence guarantees, together with a controlled gap between $L_q$ and $L$. The two viewpoints are therefore complementary rather than overlapping. We have added a discussion of [1] in Appendix B (Related Works) to make this relationship clear in the revised manuscript.

---

> ### Author Response · Authors · 2025-11-28
>
> Dear Reviewer fvbi,
>
> I hope this message finds you well. We wanted to gently remind you that the deadline for the discussion phase is approaching, and we would greatly appreciate it if you could take a moment to review our responses.
>
> Your feedback is very valuable to us, and we wanted to ensure all your concerns have been addressed satisfactorily. If there are any additional concerns or points of clarification, we are more than happy to address them.
>
> Thank you for your time and consideration.
>
> Best wishes,
>
> Authors

---

> ### Author Response · Authors · 2025-12-02
> **Theoretical proof to Minor Question1**
>
> We thank the reviewer again for raising this point. In the revised manuscript, we have added a **complete theoretical proof** showing that  $\sum_{j=1}^n\gamma_j /q = 1$ for general $(n,s,q)$. The proof relies on an induction argument together with a standard combinatorial identity obtained by grouping subsets according to the position of their $(m+1)$-th largest element. This derivation is now included in **Appendix D.2**, in addition to the numerical validations already provided.

---

### Official Review · Reviewer_TnEE · 2025-10-30

**Soundness:** 4
**Presentation:** 3
**Contribution:** 3
**Rating:** 4
**Confidence:** 2

**Summary:**

This paper presents OrderDP, a novel framework for dynamic data pruning aimed at addressing the biased gradients and training instability inherent in existing methods. The core idea is a two-stage selection process: uniformly sampling a candidate pool, then selecting the top-q highest-loss examples for the gradient update. The key innovation is theoretical: rather than fixing the bias for the original objective, the authors introduce a surrogate loss function and prove that their method provides an unbiased gradient estimate for this new objective. This allows them to establish formal convergence and generalization guarantees. Empirically, OrderDP demonstrates superior performance and stability on CIFAR and ImageNet benchmarks, achieving significant training speed-ups with minimal to no accuracy loss.

**Strengths:**

1.	The paper introduces a novel two-stage sampling method that combines uniform random sampling for exploration with a top-q, high-loss selection for exploitation. This straightforward design effectively balances the need for data diversity with the selection of informative samples and is simple to integrate into existing training pipelines.

2.	Theoretical Analysis The work is supported by a relatively thorough theoretical analysis, which is a key differentiator from many heuristic-based approaches. By introducing a surrogate loss function, the authors reframe the biased pruning problem into an unbiased optimization of a new objective. This allows them to establish formal convergence rates and generalization bounds.

**Weaknesses:**

Major concern:
1.	The authors motivate their work by highlighting the critical issue of biased gradient estimation in existing dynamic data pruning methods, where the bias is defined with respect to the full-dataset empirical risk, L(θ). However, the proposed OrderDP method does not directly yield an unbiased estimator for the gradient of L(θ). Instead, it cleverly reframes the problem by introducing a surrogate loss, L_q(θ), for which the computed gradient is indeed an unbiased estimator.This constitutes a subtle but significant mismatch between the initial problem formulation and the solution. The central premise of the paper shifts from 'fixing the bias for the original objective' to 'optimizing a different, tractable objective'. The validity of this entire approach hinges on the crucial assumption that minimizing L_q(θ) is a sufficiently good proxy for minimizing L(θ).While the authors attempt to bridge this gap through the generalization analysis in Theorem 4, which bounds the bias between the two objectives, the paper would be strengthened by a more direct discussion of this methodological pivot.

2.	The authors make a strong claim of achieving 'loss-less' performance, which is central to the paper's positioning. The empirical results, however, suggest that this claim holds under specific, moderate pruning ratios (e.g., 30% on CIFAR-10) but not universally across more aggressive pruning settings. This discrepancy between the strong, general claim and the nuanced, conditional results could be misleading. The authors should consider refining their claim to be more precise, perhaps by using terminology like 'near-loss-less' or by explicitly stating the conditions under which true 'loss-less' performance is achieved. This would align the paper's claims more accurately with its empirical evidence.

**Questions:**

1. Can the authors clarify how minimizing the surrogate loss $L_q(\theta)$ reliably approximates minimizing the original objective $L(\theta)$? Are there empirical results or theoretical analyses demonstrating when this approximation may fail?

2. Would the authors consider explicitly discussing the methodological shift from bias correction for $L(\theta)$ to optimization under $L_q(\theta)$, and how this affects the paper’s claimed contributions?

3. Regarding the ``loss-less'' claim, could the authors specify the exact pruning conditions under which this property holds, and whether ``near-loss-less'' might be a more accurate description?

---

> ### Author Response · Authors · 2025-11-21
> **Response to Weakness**
>
> > **W1. Dynamic pruning alters the weighting in $L(\theta)$, while OrderDP is unbiased only for $L_q(\theta)$, suggesting a discussion of objective shift between the stated goal and the optimized objective.**
>
> Thank you for this valuable suggestion. Regarding the relationship between the original objective $L(\theta)$ and the surrogate objective $L_q(\theta)$, we agree that a clearer explanation is needed.
>
> We first clarify that we never claim OrderDP provides an unbiased gradient estimator for $L(\theta)$. In fact, once dynamic pruning is used to select samples, it inevitably introduces bias relative to the original uniform objective. Concretely, dynamic pruning changes the per-sample weighting from the uniform formulation $L(\theta)=\frac{1}{n}\sum_{i=1}^n L_i(\theta,z_i)$ to a non-uniform objective $L_q(\theta)=\frac{1}{q}\sum_{j=1}^n \gamma_j\,L_{(j)}(\theta,z_i)$, where we further show that the weights $\gamma_j$ follow an asymptotic Beta-like distribution. Under such non-uniform weighting, maintaining an unbiased gradient estimator for $L(\theta)$ is theoretically and practically difficult.
>
> Our goal is therefore not to “fix the bias of $L$”, but to make explicit the *implicit reweighted objective* that any dynamic pruning method inherently optimizes, and to express this objective in a precise mathematical form (Theorem 1). The contribution of OrderDP lies in explicitly formulating this surrogate objective $L_q$, proving that its gradient estimator is strictly unbiased, and establishing convergence and generalization guarantees for it. In essence, we quantify the bias induced by dynamic pruning as a distribution shift, thereby characterizing how pruning ratio affects optimization behavior and final performance.
>
> Regarding why minimizing $L_q$ is a good approximation to minimizing $L$, Theorem 4 provides an explicit upper bound on the discrepancy between the two objectives. The approximation quality is controlled by how far the induced weights $\gamma_j$ deviate from the uniform distribution—captured by the terms $C_s$ and $Q_n(\theta;s,q)$. When the pruning ratio is high (i.e., exploitation size $q$ or the exploration size $s$ is small since $q \leq s$), the range of distribution $\gamma_j$ becomes large, producing a significant deviation from uniform distribution (range = 0) and thus a weaker approximation. As the pruning ratio decreases (i.e., $q \to s$), the spread of $\gamma_j$ shrinks, the deviation from uniform distribution becomes smaller, and both $C_s$ and $Q_n(\theta;s,q)$ decrease, improving the approximation. In the special case $q=s$, OrderDP reduces exactly to SGD, the bias term vanishes, and $L_q=L$. We visualize the distribution shift in Figure 6,7 of the revised version.
>
> Our experimental results in Section 5 match the theoretical prediction well: as the pruning ratio decreases, the accuracy drop also consistently shrinks. We also acknowledge the limits of this approximation. When the pruning ratio is extremely high or the candidate pool is very small (e.g., $q\ll s \ll n$ ), the induced weights $\gamma_j$ become highly non-uniform, the bias term in Theorem 4 grows, and the accuracy drop becomes noticeable—fully consistent with our theoretical analysis and empirical observations. We have added the discussion in **Section 3.2** in the revised manuscript.
>
> > **W2. The “loss-less” claim may be overstated and should be qualified by the pruning regimes where it truly holds.**
>
>
> Thank you for this valuable suggestion. We fully agree that using the term “loss-less” without definition may unintentionally give the impression that it holds under all pruning ratios. Indeed, no dynamic pruning method can maintain strictly zero accuracy drop under extremely aggressive pruning in practice [1].
>
> In line with this suggestion, we have already revised the wording in the Introduction to adopt a more precise and accurate description. Specifically, we now use $\text{near-lossless}$ pruning and further add a clarifying footnote in Introduction:
> “Here, near-lossless means matching full-data accuracy up to normal stochastic fluctuations (typically within 0.1\%) while achieving a noticeable training speedup.”
>
> We appreciate the reviewer’s recommendation and will continue using this more precise terminology to ensure that our claims remain fully aligned with the empirical evidence.
>
> [1] Ayed, Fadhel, and Soufiane Hayou. Data pruning and neural scaling laws: fundamental limitations of score-based algorithms.

---

> ### Author Response · Authors · 2025-11-21
> **Response to Questions**
>
> > **Q1. How does minimizing the surrogate loss $L_q$ approximate minimizing the original objective $L$? When can this approximation fail?**
>
> Thank you for this question. As we have throughly discussed the approximation in the response to W1, dynamic pruning inevitably changes the sampling distribution over training examples, so any score-based strategy introduces bias with respect to the original empirical risk $L(\theta)$. Our approach is to explicitly characterize the *implicitly reweighted objective* induced by pruning and write it in closed form as $L_q(\theta)$. Theorem 4 shows that the discrepancy between $L$ and $L_q$ can be quantified by the difference between the induced sampling distribution and the uniform one: when $q$ (or a small exploration size $s$ since $q \le s$) is small, the range of the distribution $\gamma_j$ over $j \in \{1,\dots,n\}$ is large (while uniform weights have zero range), leading to a large bias and a poor approximation; as the pruning ratio decreases (i.e., $q \to s$), the range of $\gamma_j$ shrinks, the bias relative to uniform weights diminishes, the terms $C_s$ and $Q_n(\theta;s,q)$ in the bound become smaller, and the approximation improves. In the special case $q = s$, OrderDP degenerates to standard SGD, the bias term is zero, and $L_q = L$. This behavior is illustrated in Appendix Fig. 6,7, and our experiments show the same trend: as the pruning ratio decreases, the accuracy drop also decreases.
>
> > **Q2. Should the methodological shift from correcting the bias of $L$ to optimizing the surrogate objective $L_q$ be discussed more explicitly, and how does this affect the paper’s contributions?**
>
> Thank you for this helpful suggestion. As we have discussed the methodological shift in the response to W1, once dynamic pruning is used to select samples and discard samples, it inevitably introduces bias relative to the original uniform objective $L(\theta)$. In this sense, reframing the bias as optimization of an implicitly reweighted objective is indeed methodologically important and worth stating clearly.
>
> To clarify, this “shift” is not a change of goal but rather a clarification of the inherent mechanism of dynamic pruning. All dynamic pruning methods implicitly optimize a reweighted objective, though this has typically remained implicit in prior work. The theoretical contribution of OrderDP is precisely that it **makes this implicit objective explicit** by formulating it as the analytically tractable surrogate $L_q$, proving that the gradient estimator is unbiased for $L_q$, and establishing convergence and generalization guarantees. Thus, the relationship between $L$ and $L_q$ does not weaken our contribution—on the contrary, it strengthens it by providing a rigorously defined surrogate objective and a distribution-shift–based analysis that quantifies the bias between the two objectives.
>
> We have added the discussion in Section 3.2, and we believe the reviewer’s comment underscores the value of our theoretical framework: OrderDP not only proposes a practical dynamic pruning algorithm, but also offers a unifying explanatory structure and verifiable theory for the entire class of dynamic pruning methods.
>
> >**Q3. Clarity of the term "lossless"**
>
> We are very grateful for the reviewer's suggestion. As we responded to Weakness2, we have also updated the terminology throughout the paper to ensure consistent and unambiguous usage.

---

> ### Author Response · Authors · 2025-11-28
>
> Dear Reviewer TnEE,
>
> I hope this message finds you well. We wanted to gently remind you that the deadline for the discussion phase is approaching, and we would greatly appreciate it if you could take a moment to review our responses.
>
> Your feedback is very valuable to us, and we wanted to ensure all your concerns have been addressed satisfactorily. If there are any additional concerns or points of clarification, we are more than happy to address them.
>
> Thank you for your time and consideration.
>
> Best wishes,
>
> Authors

---

### Official Review · Reviewer_S9Z2 · 2025-11-02

**Soundness:** 3
**Presentation:** 3
**Contribution:** 2
**Rating:** 4
**Confidence:** 4

**Summary:**

The paper proposes a dynamic data pruning framework, termed OrderedDP, to speed up the model training process by reducing the number of samples needed for each epoch. The author proposes to use the loss to measure the importance of the data sample together with a ranking method, where a larger loss indicates a higher demand for retention. The experiments were conducted on CIFAR-10/100 and ImageNet-1k using ResNet-50 as the model backbone. The authors have done some analysis on the generalization of the proposed method.

**Strengths:**

1. The motivation of the paper is well justified. Also, the authors try to give some theoretical analysis of the proposed method to justify its generalization capability.

2. The paper is properly written and it is not hard to follow the paper's story.

**Weaknesses:**

1. The selected backbone is not strong enough. The reproduced accuracy of the ResNet-50 backbone is lower than the model trained with a stronger training recipe and data augmentation strategy. Actually, it is important to justify that the proposed method is compatible with state-of-the-art training recipes. It is not clear if the "unimportant samples" are indeed not that important with a stronger data augmentation strategy and learning schedule.

2. The experiments are only verified on classification tasks. It is unclear whether the proposed method can be applied to other tasks, such as segmentation, detection, or generative tasks.

3. The method is only verified on CNN-based networks (ResNet-18/50), which typically show faster convergence speed over transformer-based models. It is also not clear how the proposed method performs on transformer-based model architectures.

**Questions:**

Please refer to the weakness session.

---

> ### Author Response · Authors · 2025-11-21
> **Response to Weakness1 and 2**
>
> >**W1. Concern about backbone strength and compatibility with stronger data augmentation / training recipes**
>
>
> Thank you for raising this point. We fully agree that verifying the compatibility of *OrderDP* with stronger augmentation strategies and training recipes is important for assessing its practical value.
>
> Indeed, OrderDP is independent of the data augmentation, model, and learning schedule.
> We would first like to clarify that the goal of *OrderDP* is not to define samples as absolutely “important” or “unimportant.” In data pruning, our objective is pragmatic: given a training pipeline and compute budget, how can we accelerate training without sacrificing final accuracy? Under this perspective, sample importance is inherently *relative* and varies across models, training recipes, training stages, data augmentation strategy and so on. *OrderDP* only condisers the per-sample loss as the measure of each example’s contribution to the current update, without introducing high extra computational cost.
>
> Crucially, *OrderDP* always selects data after data augmentation has been applied. If a sample becomes harder under stronger augmentation, its loss increases and it will automatically move into the top-q set and be retained. Thus, *OrderDP* responds to *current contribution under the present recipe* and does not conflict with stronger augmentation pipelines. Moreover, *OrderDP* is orthogonal to model architecture, augmentation strategies, regularization, and optimizer choice.
>
> To directly evaluate this, we conducted experiments in **Section 5.4 Sensitivity Analysis** using the Timm ImageNet training stack, which includes mixed-precision training and strong augmentation (Mixup + CutMix). The results show that *OrderDP* remains lossless under this stronger recipe:
>
> | Setting                  | Acc. |
> |--------------------------|-----------:|
> | Full data (Original)     | 76.4       |
> | Full data (Timm recipe)  | 78.4       |
> | OrderDP (30% prune,Timm recipe)      | 78.3       |
>
>
> These findings confirm that the loss ordering remains stable under stronger augmentation and that samples identified as “low-contribution” under such settings have negligible impact on final performance. We will include these results and clarifications in the revised version.
>
>
> >**W2. Concern about applicability beyond classification (segmentation, detection, generative tasks)**
>
> Thank you for raising this point. We emphasize that OrderDP is inherently **task-agnostic**: it relies only on per-sample loss as data score, which are naturally available across supervised learning tasks. Since  *OrderDP* does not assume any task-specific structure, it can be inserted into existing training loops in a fully **plug-and-play** manner without modifying the model architecture or loss function.
>
> *(A) Generative modeling — Latent Diffusion (FFHQ)*
> We follow the same setup as InfoBatch and apply OrderDP to the training of Latent Diffusion (Rombach et al., 2021) on FFHQ. Under this setting, InfoBatch achieves a 22% reduction in pretraining cost with FID 7.72 (vs. 7.53 for the original model). Under a slightly more aggressive pruning level, OrderDP achieves a 28% reduction in pretraining cost with FID 7.68.
>
> | Method     | Prune Ratio | FID |
> |------------|-------------|------:|
> | Original   | 0%          | 7.53  |
> | InfoBatch  | 22%         | 7.72  |
> | **OrderDP** | **28%**     | **7.68** |
>
> OrderDP preserves generative quality while allowing higher pruning than InfoBatch.
>
> *(B) Semantic segmentation — ADE20K (UperNet-ResNet50)*
>
> To further assess the generality of OrderDP beyond classification, we follow InfoBatch and apply our method to the semantic segmentation task. Concretely, we train UperNet–ResNet50 on ADE20K using the mmsegmentation framework. Under this setting, InfoBatch (without annealing) actually uses 65% of the original iterations (80k steps in total) to achieve lossless performance (mIoU: 40.64 vs. 40.7 original), while OrderDP attains a comparable mIoU of 40.66 with 58% of the original iterations. This shows that OrderDP extends naturally to dense prediction and can prune even more aggressively on ADE20K without degrading segmentation quality.
>
> | Method      | Iter Ratio |    mIoU |
> | ----------- | ---------: | --------: |
> | Original    |       100% |      40.7 |
> | InfoBatch   |        65% |     40.64 |
> | **OrderDP** |    **58%** | **40.66** |
>
>
> These results demonstrate that OrderDP generalizes naturally to segmentation and generative tasks, maintaining lossless behavior comparable to, or slightly better than, InfoBatch while supporting higher pruning ratios. We will highlight these findings clearly in the revised version.

---

> ### Author Response · Authors · 2025-11-21
> **Response to Weakness3**
>
> >**W3. Concern about missing results on Transformer architectures**
>
> Thank you for this helpful comment. We appreciate the concern about architectural generality. Conceptually, the core mechanism of *OrderDP* depends only on the instantaneous per-sample loss ranking and does not make any assumption about the network architecture, so it is in principle applicable to both CNNs and Vision Transformers.
>
> To verify this empirically, we have added experiments on ImageNet-1K using Swin-Tiny and ViT-Base (MAE) under the same training setup as InfoBatch. The new results are reported in **Section 5.4 (Sensitivity Analysis), Table 5**. Under this setting, we increase the prune ratio to 22.1% on Swin-T and 30.8% on ViT-B (MAE), which is higher than the ratios used in the original InfoBatch experiments. Even at these more aggressive pruning levels, *OrderDP* maintains **near-lossless** accuracy on both backbones (deviations within at most 0.1%).
>
> These results support two takeaways: (1) *OrderDP* can be directly applied to mainstream Vision Transformer architectures **without any modification** to the model or training pipeline, and (2) even at higher prune ratios than those previously reported, Transformer models retain essentially the same accuracy, indicating that our loss-based ordered sampling is stable and effective beyond CNNs. We will highlight these transformer results more clearly in the revised version to address this concern.

---

> ### Author Response · Authors · 2025-11-28
>
> Dear Reviewer S9Z2,
>
> I hope this message finds you well. We wanted to gently remind you that the deadline for the discussion phase is approaching, and we would greatly appreciate it if you could take a moment to review our responses.
>
> Your feedback is very valuable to us, and we wanted to ensure all your concerns have been addressed satisfactorily. If there are any additional concerns or points of clarification, we are more than happy to address them.
>
> Thank you for your time and consideration.
>
> Best wishes,
>
> Authors

---

### Author Response · Authors · 2025-12-02
**General Response to the Area Chair**

Dear AC, SAC, and PC,

We would like to thank all reviewers for their careful assessments and constructive critiques, which are invaluable for improving our manuscript. We are encouraged that reviewers consistently highlight three key positive aspects of our work, namely (i) the conceptual novelty and principled formulation of addressing gradient bias in dynamic data pruning (Reviewers `S9Z2`, `TnEE`, `fvbi`), (ii) the strength and depth of our theoretical analysis, which all reviewers recognized as a central contribution of the work, including unbiased gradients for the surrogate loss and rigorous convergence and generalization guarantees (Reviewers `S9Z2`, `TnEE`, `fvbi`, `umfZ`), and (iii) the competitiveness and robustness of our empirical evaluation across datasets, optimizers, and challenging high-compression regimes (Reviewers `S9Z2`, `fvbi`, `umfZ`). We sincerely appreciate these positive assessments.


**1. Response to Key Reviewer Concerns**

Since the reviewers raised largely non-overlapping concerns, we summarize the main issues into three themes and briefly describe how they have been addressed in our rebuttal.

**(a) Relation between the surrogate objective $\mathcal{L}_q$ and the original loss $\mathcal{L}$.**

Reviewer `TnEE` requested clearer justification for the methodological shift from correcting bias with respect to $\mathcal{L}$ toward unbiased optimization of $\mathcal{L}_q$. In our rebuttal, we clarified the motivation for introducing the surrogate loss, explained why minimizing $\mathcal{L}_q$ is a principled and tightly coupled proxy for minimizing $\mathcal{L}$, and highlighted the theoretical guarantees in Theorem 4 that uniformly bound the discrepancy between the two objectives. We also discussed when this approximation may degrade and provided empirical evidence. We have added this discussion in the revised manuscript.

**(b) Applicability across architectures, tasks, and training pipelines.**

Reviewer `S9Z2` asked whether OrderDP remains effective under stronger training recipes, non-classification tasks, and architectures beyond CNNs. In our rebuttal, we emphasized that OrderDP is a fully plug-and-play framework that operates solely on per-sample loss after augmentation and is therefore agnostic to model architecture, task type, and training schedule. We further demonstrated this by presenting additional experiments under *stronger ImageNet pipelines*, *Vision Transformer architectures*, and tasks such as *generative modeling* and *semantic segmentation*. These extensions consistently exhibit near-lossless performance and support the broad applicability of OrderDP across models, augmentation strategies, and tasks. We have included the corresponding explanations and updates in the revised manuscript.

**(c) Clarifications of definitions, assumptions, and implementation details.**

Reviewers `fvbi` and `umfZ` requested clarifications regarding mini-batch implementation, compression ratios, dataset coverage, noisy-label behavior, and baseline definitions. These concerns have been fully addressed in the rebuttal and resolved in the revised manuscript through refined definitions, clearer explanations, and the added proof of $\sum_j \gamma_j / q = 1$.


**2. Response to Reviewer Suggestions**

All remaining suggestions from the reviewers, including terminology refinement, clarification of optimizer compatibility, and presentation improvements, have been fully incorporated into the revised manuscript.

**3. On Reviewer Follow-Up**

Due to the system incident affecting this rebuttal cycle, we unfortunately did not receive follow-up questions or updated comments from the reviewers after submitting our detailed responses. Nevertheless, we believe that our rebuttal, together with the additional experiments and theoretical clarifications, thoroughly addresses all concerns, both technically and empirically.

We respectfully invite the Area Chair to evaluate our submission in light of the rebuttal and the additional results provided. We believe that the revised version presents a technically solid, empirically validated, and practically relevant contribution to dynamic data pruning.

Thank you very much for your time and consideration.

Authors

---

### Meta-Review · Area_Chair_BKjz · 2025-12-21

**Summary:**

This paper proposed a novel plug-and-play dynamic data pruning framework, OrderDP, that can significantly reduce the required training data size while ensuring theoretically guanranteed near-lossless model performance.

The reviewers raised multiple core concerns:
(1)Theoretical guarantee and the surrogate objective.
Validity of the "unbiased training" guarantee (surrogate loss vs. original loss) and the generalization bounds.
(2)Experimental design, including absent SOTA dynamic pruning methods, limited model diversity (limited to ResNets only), and limited pruning ratios (e.g., 20%–60%).
(3)The over-claim of achieving 'loss-less' performance.

The authors’s rebuttal have duly addressed most concerns accordingly. Minor outstanding concerns, including the experimental scope and the overclaim, are non-critical for the paper’s main contributions.

In summary, given the theoretical contribution and empirical success of this work, I recommend the submission for Accept (Poster).

**Reviewer Concerns:**

I think the main concerns have been partially addressed by the authors’ rebuttal.

(1) Theoretical guarantee and the surrogate objective:
- The authors highlighted the theoretical guarantees in Theorem 4 that uniformly bound the discrepancy between the two objectives. They also discussed when this approximation may degrade and provided empirical evidence.

(2) Experimental design, including absent SOTA data pruning methods and limited model diversity (limited to ResNets only).
- The authors provided 5 recent data pruning baselines, demonstrating OrderDP’s superior accuracy-stability tradeoffs and faster runtime.
- The experiments are generalized to ViTs and MobileNets, across diverse model architectures.
- The authors provided clarifications regarding mini-batch implementation, compression ratios, dataset coverage, noisy-label behavior, and baseline definitions.

I think two remaining concerns are still outstanding.

(1) As LLMs become increasingly important, the additional experiments on LLMs/autoregressive models are still missing in this work.

(2) Nearly all reviewers believe that the claim of achieving 'loss-less' performance is too strong, theoretically and empirically. This claim only holds under relatively strict conditions. It would be better to reorganize the relevant statements and avoid the overclaim in the revision.

**Reviewer Scores:**

Reviewer S9Z2: 4 -> 6
- Justification: I think the authors’ rebuttal has addressed W1 and W2 partially. The outstanding concern is lacking LLM/autoregressive modeling experiments.

Reviewer TnEE: 4 -> 6
- Justification: I think the authors’ rebuttal has addressed W1 but not W2. The outstanding concern is the overclaim of achieving 'loss-less' performance.

Reviewer fvbi: 8 -> 8
- Justification: The authors’ rebuttal has addressed W1 and W2. I see no outstanding concern for this reviewer.

Reviewer umfZ: 6 -> 8
- Justification: The authors’ rebuttal has addressed W1 and W2. I see no outstanding concern for this reviewer.

---

### Decision · Program_Chairs · 2026-01-26

Accept (Poster)